Resource

# Reduced protein-coding transcript diversity in severe dengue emphasises the role of alternative splicing

Priyanka Mehta[1,2,*] ⓘ, Chinky Shiu Chen Liu[1,*], Sristi Sinha[1], Ramakant Mohite[1], Smriti Arora[1], Partha Chattopadhyay[1,2], Sandeep Budhiraja[3], Bansidhar Tarai[3], Rajesh Pandey[1,2] ⓘ

Dengue fever, a neglected tropical arboviral disease, has emerged as a global health concern in the past decade. Necessitating a nuanced comprehension of the intricate dynamics of host–virus interactions influencing disease severity, we analysed transcriptomic patterns using bulk RNA-seq from 112 age- and gender-matched NS1 antigen–confirmed hospital-admitted dengue patients with varying severity. Severe cases exhibited reduced platelet count, increased lymphocytosis, and neutropenia, indicating a dysregulated immune response. Using bulk RNA-seq, our analysis revealed a minimal overlap between the differentially expressed gene and transcript isoform, with a distinct expression pattern across the disease severity. Severe patients showed enrichment in retained intron and nonsense-mediated decay transcript biotypes, suggesting altered splicing efficiency. Furthermore, an up-regulated programmed cell death, a haemolytic response, and an impaired interferon and antiviral response at the transcript level were observed. We also identified the potential involvement of the *RBM39* gene among others in the innate immune response during dengue viral pathogenesis, warranting further investigation. These findings provide valuable insights into potential therapeutic targets, underscoring the importance of exploring transcriptomic landscapes between different disease sub-phenotypes in infectious diseases.

## Introduction

Dengue, a pervasive vector-borne viral disease, annually afflicts around 400 million individuals and claims the lives of 40,000 people worldwide (1). Despite witnessing a substantial increase in dengue cases over the past decade, it remains classified as a neglected tropical disease of significant global concern (2, 3). In India alone, the National Centre for Vector Borne Disease Control has reported over 820,000 cases and 1,100 deaths since 2018 (4).

Infection by any of the four major circulating serotypes, DENV1-4, leads to frequent seasonal outbreaks and epidemics, giving rise to evolving strains and severe disease manifestations.

According to the 2009 WHO classification, dengue fever is clinically categorised into dengue without and with warning signs and severe dengue (5). Patients without warning signs typically present with the classic symptoms of dengue fever, including leukopenia, nausea, and vomiting. On the contrary, patients with warning signs display additional symptoms that raise concerns about the possibility of progressing to severe dengue. These warning signs may include plasma leakage, resulting in haemoconcentration, elevated haematocrit, pleural effusion, and ascites (6, 7, 8). A dramatic reduction in blood pressure can lead to hypovolaemic shock and even death. Severe thrombocytopenia, marked by a significant reduction in blood platelet counts, can result in dengue haemorrhagic shock (DSS) in about one-third of severe dengue cases (9, 10). Dengue fever is frequently associated with an elevated risk of severe secondary infections involving a different DENV serotype, known as antibody-dependent enhancement. The protective immunity against DENV lasts for a limited time window, after which pre-existing IgG antibodies targeting the primary DENV serotype cross-react with the invading new DENV serotype. However, this cross-reactivity does not lead to neutralisation of the invading virus. This phenomenon heightens the risk of severe infection, as the existing cross-reactivity enhances viral entry into Fc receptor–bearing cells resulting in exacerbated viraemia and uncontrolled infection (11).

The DENV genome, like other RNA viruses, encodes a limited set of proteins and relies on the host machinery for replication (12). During infection, viral components manipulate cellular processes, modify intracellular membranes, alter host metabolic pathways, and suppress innate antiviral responses (13). The virus strategically targets the host spliceosomal machinery, redirecting it to express viral proteins (14). Studies have highlighted the takeover of host spliceosome by the virus, contributing to the preferential expression of viral proteins and unhindered DENV replication. De Maio

[1]Division of Immunology and Infectious Disease Biology, INtegrative GENomics of HOst-PathogEn (INGEN-HOPE) Laboratory, CSIR-Institute of Genomics and Integrative Biology (CSIR-IGIB), Delhi, India [2]Academy of Scientific and Innovative Research (AcSIR), Ghaziabad, India [3]Max Super Speciality Hospital (A Unit of Devki Devi Foundation), Max Healthcare, Delhi, India

Correspondence: rajesh.p@igib.res.in, rajeshp@igib.in
*Priyanka Mehta and Chinky Shiu Chen Liu contributed equally to this work

et al demonstrated that the dengue RNA-dependent RNA polymerase (RdRp), NS5 protein, interacts with host core components of the U5 snRNP particle, CD2BP2 and DDX23, potentially rewiring host gene alternative splicing (15). In short, alternative splicing involves the generation of multiple transcript isoforms from a single gene through various processes, such as exon skipping, intron retention, and alternative splice site usage. This intricate molecular phenomenon adds another layer of diversity to the transcriptome, potentially influencing the functional repertoire of the genes involved. This may lead to the down-regulation of host antiviral factors, allowing unhindered DENV replication. Another study by Pozzi et al showed that DENV NS5 targets the host splicing factor RBM10, marking it for proteasomal degradation (16). The loss of RBM10 impairs the production of interferons and proinflammatory cytokines essential for DENV restriction. Modulation of the host splicing machinery is not only restricted to DENV infection, but rather, it is a universal strategy adopted by viruses such as influenza A and SARS-CoV-2 to facilitate viral replication and dissemination (17, 18, 19). These findings underscore the significance of substantial remodelling of the host transcriptome's splicing landscape strategically driven by the invading virus to counteract or mitigate transcriptomic alterations associated with the host defensive response.

Therefore, gaining insights into the dynamic changes within the host transcriptome across varying degrees of dengue disease severity is crucial for a comprehensive understanding of disease pathogenesis (20, 21, 22). This elucidation holds the key to reducing fatality rates and developing cost-effective strategies to counter the rising epidemiological trends of dengue. This is of immense value in the tropical regions of the world, especially South Asia, in countries such as India, Bangladesh, Indonesia, Singapore, Malaysia, and Vietnam (23, 24, 25). Integrative studies that combine clinical and host genome-level response (transcriptomic) are imperative in this context (26).

Thus, in this study, we employed whole-transcriptome bulk RNA sequencing of 112 clinical blood samples, from NS1 antigen–positive hospital-admitted dengue patients, focusing on the host's transcriptome diversity across different clinical severity sub-phenotypes. Although in general transcriptome-based studies derive conclusions from overall gene expression, our approach delves into the potential of distinct *transcript isoform expression landscapes* as pivotal regulators of dengue severity. To this end, we complemented gene-level differential expression (DE) analysis with transcript-level DE analysis to identify severity-specific alterations in the diversity of differentially expressed transcript isoforms. Subsequently, we examined the global severity-specific alterations in splicing patterns because of DENV infection. Finally, we delved into understanding how differential isoforms contribute to overall gene expression and their functional implications by examining the patterns of differential transcript usage (DTU) across severity.

# Results

## Clinical characterisation and classification of dengue patient cohort

In this study, 112 NS1 antigen–positive hospital-admitted primary dengue infection patients were enrolled to explore the host transcriptome dynamics and alternative splicing to elucidate their potential role in modulating disease severity. Blood samples were collected from these patients, and total RNA was isolated for a comprehensive assessment of the whole transcriptome (Fig 1A). The patients were stratified into three severity sub-phenotypes based on the 2009 WHO classification—*dengue without warning signs* referred to as mild (n = 45), and *dengue with warning signs* that was further sub-grouped into moderate (n = 46) and severe (n = 21); based on the presence of two key clinical parameters of leukopenia and thrombocytopenia (5). Using the bulk RNA-sequencing data, we performed both gene-level and transcript-level differential expression analyses between *severe versus mild*, *severe versus moderate*, and *moderate versus mild* patient groups, followed by differential splicing analysis and differential transcript isoform usage analysis to investigate factors modulating dengue severity sub-phenotypes.

Upon assessing the clinical parameters associated with dengue-infected patients, we observed no statistically significant differences in age and gender between the severity sub-groups (Table 1 and Fig 1B). This ensured the absence of age- and gender-related potential biases in subsequent analyses. Severe thrombocytopenia and leukopenia were observed in the severe dengue patients, whereas moderate patients showed intermediate platelet counts but severe leukopenia compared with the mild patients (Fig 1C and D). Notably, we also observed an increase in lymphocyte count in moderate and severe conditions (Fig 1E). Moderate and severe forms of dengue were also characterised by neutropenia (Fig 1F). We also observed high bilirubin and serum glutamic–oxaloacetic transaminase levels in the severe dengue (Fig 1G), which are indicators of liver damage. The evaluated clinical parameters thus re-affirm the severity-based classification of the dengue patients. Other symptoms of dengue, including vomiting, diarrhoea, fever, and headache, were, however, not significantly different across the patients of varying severity (Fig 1H). Although study samples were NS1 antigen–positive for dengue, we also tried to identify dengue serotype(s) for these patients. Towards that, we captured viral reads from the bulk RNA-seq data using the non-human mapped reads. Dengue genome analysis revealed that most of the samples showed DENV2 infection (Table S1). For some of the samples, we had inadequate viral reads for serotype identification, which were distributed across the disease severity.

## Expression dynamics of genes and transcripts across severity sub-phenotypes

In response to the viral infection, the host transcriptome is rapidly altered to either promote host protective functions to limit infection-driven pathology, or augment survival and spread of viral progenies, leading to severe infection. We performed a detailed analysis of the host gene- and transcript-level differential expression, identifying events that contribute to the dynamic changes in the host transcriptome across disease severity (mild, moderate, and severe). Taking a significance threshold of $P$ adj < 0.05 and $\log_2$ fold change ≥ |1.5|, for both differential gene expression (DGE) and differential transcript expression (DTE), we identified 214 DTEs between severe and mild patients (Fig 2A and Table S2A). Among

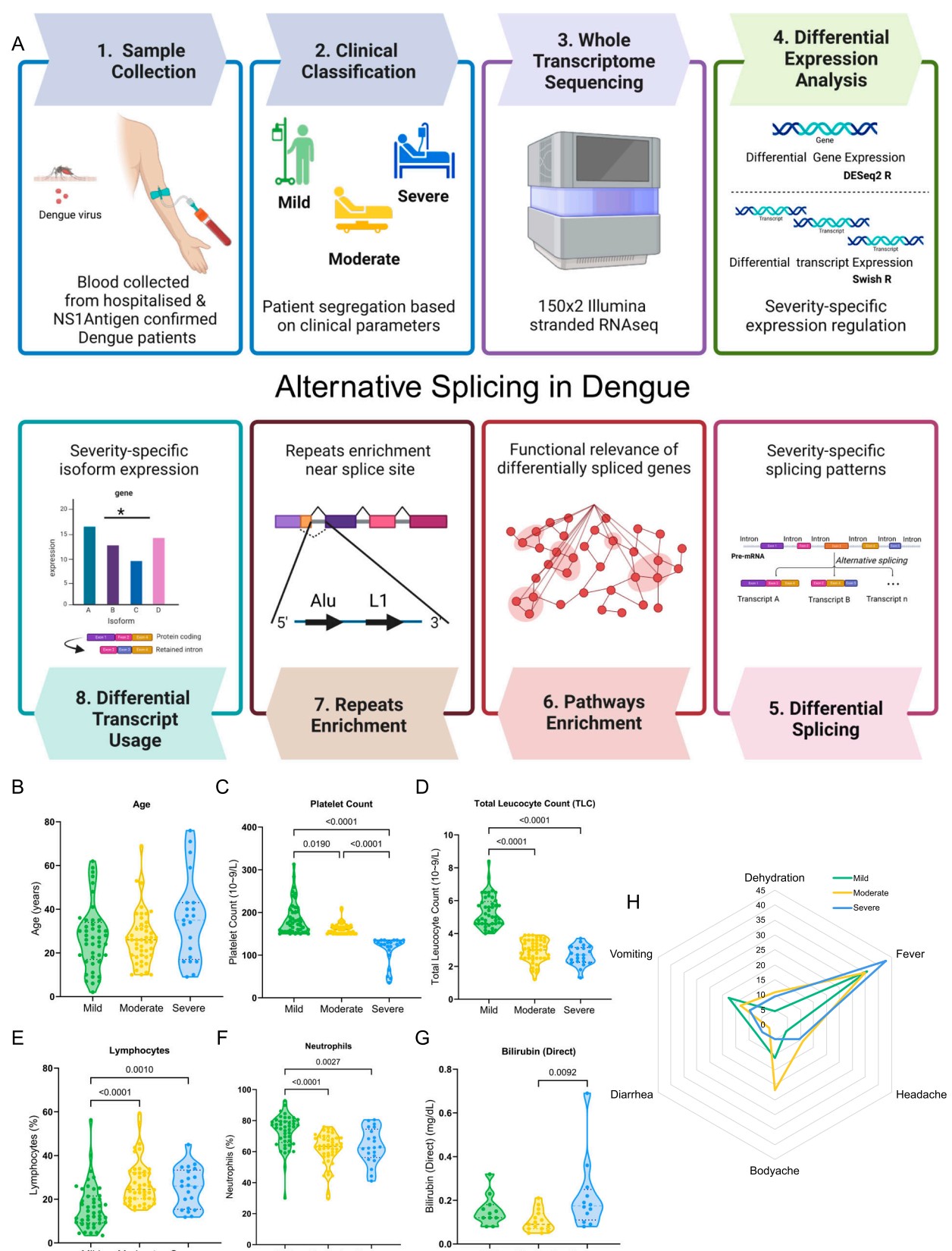

**Figure 1. Study design overview, patient stratification, and clinical characteristics of the cohort.**
**(A)** Schematic flow chart illustrates the sequential process, starting from sample collection, clinical segregation based on the clinical characteristics of transcriptome sequencing, transcript-level RNA-seq data analysis, and downstream functional analysis, and interpretation for dengue-positive patients categorised as mild, moderate,

these, 10 were down-regulated, and 204 were up-regulated in the severe patients. In the comparison between the severe and moderate cases, 11 DTEs were up-regulated, and 15 were down-regulated in the severe patients (Fig 2B and Table S2B). Interestingly, in the moderate versus mild comparison, all 122 DTEs were down-regulated in the moderate (Fig 2C and Table S2C).

In the case of DGEs, we identified 595 genes between the severe and mild patients (21 down-regulated, 574 up-regulated) (Fig S1A and Table S3A). Between severe and moderate, 37 genes were captured (2 down-regulated, 35 up-regulated) (Fig S1B and Table S3B). In the comparison between moderate and mild, 17 genes were down-regulated, whereas four were up-regulated (Fig S1C and Table S3C). There was a minimal overlap between the DGEs and DTEs across the disease severity sub-phenotypes. Notably, only 47 genes were common in the severe versus mild comparison, with three genes shared between DGEs and DTEs in both the severe versus moderate and moderate versus mild comparison groups (Fig 2D). These findings emphasise unique patterns in the expression dynamics of both genes and transcripts associated with various dengue disease severity in our study cohort.

We delved further into specific transcript features to identify potential enrichment/depletion of severity-specific isoform patterns that were significantly expressed compared with their non-significant counterparts/isoforms, referred to as differential isoforms (DIs) (Fig 2E). In contrast to the global distribution of biotypes, a notable decrease in the conventional protein-coding biotype was evident ($P = 0.03$), coupled with a simultaneous increase in non-canonical transcripts, such as long non-coding RNA (lncRNA) ($P = 0.01$) and retained intron transcript biotypes ($P < 0.001$) among severe dengue patients (Fig 2F). Protein-coding transcripts were, however, significantly enriched in both the mild ($P < 0.001$) and moderate ($P = 0.003$) patients. Although the moderate group did not show any presence of significantly expressed lncRNAs, mild patients showed significant deficiency of lncRNA transcripts ($P < 0.001$). Furthermore, transcripts destined for nonsense-mediated decay (NMD) were marginally but significantly enriched in the moderate but depleted in mild patients ($P < 0.001$), whereas we observed a significant decrease in such transcript biotypes in the severe patients ($P < 0.001$). Overall, a significant variation was observed in the proportions of transcript biotypes across the disease severity.

We performed the gene set enrichment analysis, considering pathways with $P < 0.05$ and with at least two genes as significant; we noted significant positive enrichment of immune response pathways including B-cell receptor signalling, complement activation, Fc receptor–mediated signalling, innate immune response, and infectious disease response, at both the gene and transcript isoform levels (Table S4 and Fig 2G). Interestingly, we also observed differential regulation of critical pathways exclusively at the transcript isoform level. Specifically, there was enrichment of transcript isoforms that regulate chromatin modifications and cell cycle in the severe versus mild comparison. Although there was significant positive enrichment of DTEs regulating programmed cell death (PCD) pathways in the severe versus mild condition, there was negative enrichment of DTEs affecting this pathway in the moderate versus mild condition. We also observed the ablated expression of DTEs affecting interferon signalling in severe versus mild and an antiviral response in severe versus moderate. Given that increased haemolysis is a characteristic of severe dengue, we observed enrichment of DTEs affecting haemolysis in the severe versus mild conditions.

Finally, we explored the genomic features of the DTEs across the disease severity (Fig S1D–F). We found that significant DTEs have lower numbers of exons in the severe versus mild patients, when compared to the non-significant differential isoforms (DIs) ($P < 0.001$) (Fig 2H). In addition, we noted a decrease in transcript length, particularly in the coding sequence (CDS), in the severe patients, between significant DTEs compared with the DIs ($P = 0.008$). These findings suggest a reduction in the length of transcript isoforms associated with severe dengue similar to our previous study on SARS-CoV-2 differential disease severity (Fig S1G–I) (27). The decreased number of exons observed in severe dengue aligns with the diminished length of transcript isoforms that exhibit differential expression (DTEs) in these individuals. Importantly, despite the shorter transcripts, severe dengue is characterised by greater transcript diversity, as evidenced by an increase in the ratio of differentially expressed transcript isoforms in significant DTEs when compared to the DIs ($P < 0.001$) (Fig 2I). These findings suggest that dengue severity is linked to a distinct transcriptional/post-transcriptional regulation that results in the formation of shorter yet diverse transcripts.

## Escalated differential splicing associated with dengue disease severity

In our earlier findings, we noted an increase in transcript diversity with the severity of dengue, accompanied by a reduction in the proportion of protein-coding biotypes. To understand the distribution of isoform diversity, we delved into alternative splicing, which is one of the main contributors to the heightened transcript diversity. In our assessment of splicing patterns across severity, we identified 1,040 genes with significant differential splicing between the severe and mild patients (Table S5). In contrast, there were only 141 genes with significant differential splicing events in the severe versus moderate and 235 genes in the moderate versus mild patients (Fig 3A). Despite variations in transcript expression levels being potentially influenced by splicing events, there was a limited overlap observed between DTEs, DGEs, and differentially spliced (DS) genes across the disease severity. Specifically, we noted an

and severe. **(B, C, D, E, F, G)** Sample-wise distribution of clinical parameters across the mild, moderate, and severe patients for (B) age of the patients, (C) platelet counts (in $10^9$/litre), (D) total leucocyte count (in $10^9$/litre), (E) lymphocyte count (in %), (F) neutrophil count (in %), and (G) bilirubin (direct) (in mg/dl). **(H)** Radar plot represents the clinical symptoms manifested by the patients in three severity groups. A Mann–Whitney $U$ test was used to test for significance, and only significant associations are depicted in the plot.

**Table 1. Clinical characteristics of the patient cohort.**

| Clinical parameters | Mild (n = 45) | Moderate (n = 46) | Severe (n = 21) | P-values |
|---|---|---|---|---|
| Age | 27 (18–33) | 26 (19–31.75) | 35 (18–43) | 0.14 |
| Gender F|M | 15|30 | 17|29 | 4|17 | 0.34 |
| NS1 antigen | 3.22 (0.83–3.60) | 3.5 (0.86–3.5) | 3.5 (1.9–3.5) | 0.75[a] |
| Total leucocyte count (TLC) | 5 (4.6–5.9) | 3 (2.5–3.575) | 2.7 (2.3–3.1) | **<0.001** |
| RBC count | 4.76 (4.51–4.99) | 4.835 (4.58–5.02) | 4.78 (4.51–5.58) | 0.76 |
| Packed cell, volume | 41.2 (40.1–42.8) | 42.25 (38.4–44.22) | 43.9 (40.4–46.3) | 0.18 |
| Platelet count | 175 (155–202) | 160 (150–165) | 125 (112–135) | **<0.001** |
| Neutrophils | 75.9 (66.7–79.5) | 63.3 (58.1–68.65) | 62.6 (57.3–73) | **<0.001** |
| Lymphocytes | 14.1 (9.4–21.3) | 24.5 (20.3–31.975) | 26 (15.8–32.7) | **<0.001** |
| Monocytes | 9.8 (8–12) | 10 (7.85–11.475) | 9 (6.9–11) | 0.32 |
| Eosinophils | 0.2 (0–0.725) | 0.2 (0.1–0.875) | 0.5 (0.1–1) | 0.32 |
| Basophils | 0.5 (0.3–0.6) | 0.6 (0.225–0.8) | 0.5 (0.3–0.8) | 0.69 |
| Hb | 13.6 (12.75–14.3) | 13.9 (12.52–14.5) | 14.1 (13.1–15.3) | 0.28 |
| Total protein | 7.3 (7–7.6) | 6.7 (6.5–7.3) | 6.7 (6.05–7.125) | **0.03**[a] |
| Albumin | 4.4 (4.05–4.5) | 4 (3.8–4.15) | 4 (3.675–4.1) | **0.04**[a] |
| Globulin | 3.1 (2.95–3.3) | 3 (2.65–3.15) | 2.65 (2.3–2.925) | 0.06[a] |
| A.G. ratio | 1.4 (1.3–1.55) | 1.3 (1.2–1.5) | 1.55 (1.375–1.6) | 0.19[a] |
| Bilirubin (total) | 0.7 (0.55–0.8) | 0.4 (0.3–0.55) | 0.75 (0.5–0.9) | **0.02**[a] |
| Bilirubin (direct) | 0.135 (0.12–0.17) | 0.09 (0.07–0.13) | 0.17 (0.13–0.23) | **0.02**[a] |
| Bilirubin (indirect) | 0.56 (0.43–0.615) | 0.31 (0.24–0.42) | 0.55 (0.39–0.71) | **0.03**[a] |
| SGOT- aspartate transaminase (AST) | 41 (34.5–76) | 59 (44–80.5) | 97.5 (74.25–141.5) | **0.03**[a] |
| SGPT- alanine transaminase (ALT) | 42 (20–95) | 41 (25.5–59) | 66 (51.75–80.25) | 0.13[a] |
| AST/ALT ratio | 0.93 (0.82–1.725) | 1.48 (1.13–1.95) | 1.705 (1.2–2.17) | 0.08[a] |
| Alkaline phosphatase | 67 (60–89) | 80 (59.5–100) | 82.5 (59.75–124.25) | 0.61[a] |
| GGTP (gamma GT), serum | 37 (24.5–46.5) | 19 (14–41) | 49 (32–82) | 0.08[a] |

Data represented as median (IQR) or n (%).
[a]Incomplete data points in either group.

overlap of 13 transcript isoforms between the DTEs and DS in severe versus mild, compared with only four and six overlaps in severe versus moderate and moderate versus mild, respectively. Hence, although dengue severe states are characterised by remodelling of the host transcriptome splicing landscape, it seems that these changes do not necessarily impact the levels of gene or transcript expression.

To elucidate the functional implications of differential splicing on gene function, we investigated the pathways associated with the genes undergoing differential splicing. We specifically observed the enrichment of immune response pathways in severe compared with mild conditions (Fig 3B). These pathways included TLR, nod-like receptor, and C-type lectin receptor innate recognition; neutrophil response; interferon, TNF, and interleukin cytokine response; Fc receptor–mediated response; MHC-I and MHC-II; and T-cell and B-cell signalling pathways. Molecular signalling pathways associated with spliced transcripts included NFκB, PI3-Akt, Wnt/beta-catenin, Rho GTPase, and VEGF signalling pathways. Other homeostatic functions affected by splicing changes included the cell cycle, post-translational modifications, alternate splicing,

epigenetic regulation, DNA damage response, and programmed cell death pathways. Platelet activation and erythropoietin signalling were also enriched among spliced transcripts in the severe patients. In addition, we observed significant splicing changes in the genes associated with pathogen entry into the host, endocytosis, and phagocytosis. The severe versus moderate exhibited enrichment of spliced transcripts associated with the innate immune response, along with RNA and protein modification pathways (Fig 3C). In contrast, exclusive enrichment of innate immune response pathways, such as neutrophil activation, TLR recognition, and interferon and interleukin signalling, was observed in moderate versus mild patients (Fig 3D).

We also observed severity-specific enrichment or reduction in certain splicing events in comparison with their global distribution (Fig 3E). All splicing events, except for skipped exons (SE), showed significant differences across all three groups in differentially spliced (DS) genes. The alternative 5′ splicing site (A5SS) and retained intron (RI) events exhibited a consistent reduction in the DS genes across all severity conditions in comparison with the global splicing event ($P < 0.001$ for all three groups). Interestingly, in

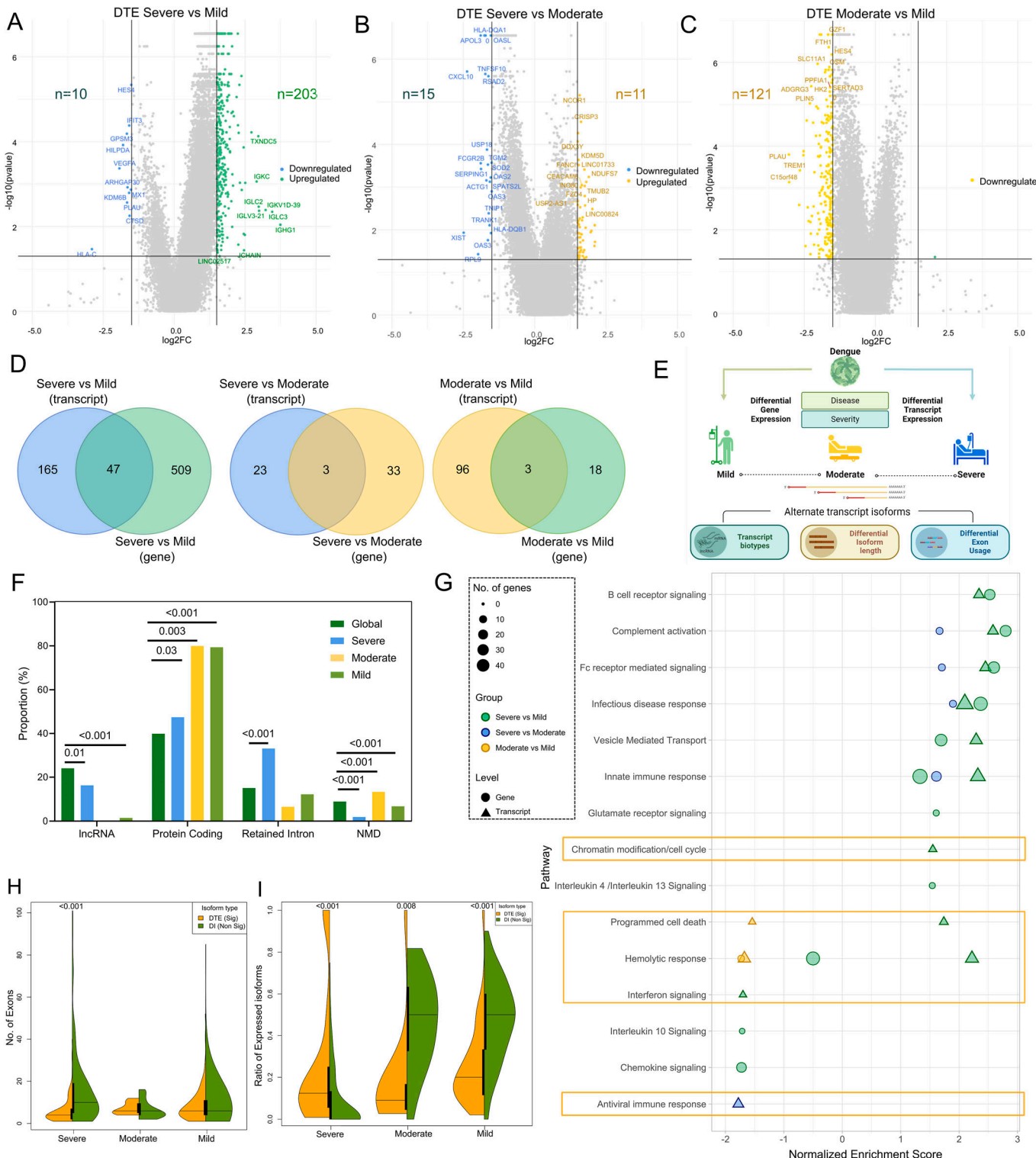

**Figure 2. Illustration of the patterns of differentially expressed transcripts across the severity sub-phenotypes and their characteristics.**
**(A, B, C)** Violin plots depict differentially expressed transcripts (DTEs) between (A) severe versus mild, (B) severe versus moderate, and (C) moderate versus mild patients. The x-axis indicates the log₂ fold change, whereas the y-axis shows the log₁₀ (*P*-values). Coloured dots denote isoforms significant in the higher severity group compared with the lower severity. **(D)** Venn diagrams showcase the overlap between significantly differentially expressed genes and transcripts across different severity groups. **(E)** Graphic visualisation characterises alternate transcript isoforms across severity groups. **(F)** Bar plot illustrates the distribution of transcript biotypes across severity groups compared with the global distribution. A chi-square test was performed to check for significance. **(G)** Combined dot plot displays groups of significantly enriched Reactome pathways for the differentially expressed gene (as circles) and DTE (as triangles), with severity comparison patient groups represented in green (severe versus mild), blue (severe versus moderate), and yellow (moderate versus mild). The size of the icons reflects the number of genes involved in the pathway. **(H)** Violin

severe DTE genes, RI was significantly higher compared with the global and DS genes ($P$ = 0.001), which aligns with the higher presence of a retained intron biotype in the severe group compared with the global and moderate/mild in our previous result (Fig 2F). In DS genes, we see a slight albeit significant reduction in the prevalence of alternate 3′ splicing site (A3SS) events ($P$ = 0.016, $P$ < 0.001, and $P$ = 0.002 in severe, moderate, and mild groups, respectively). A3SS was significantly higher in the DTEs of severe and moderate patients ($P$ < 0.001 and 0.017, respectively), but lower in mild ($P$ < 0.001), when compared to the overall global splicing events in our dataset. In addition, we observed significant enrichment of mutually exclusive exon events in the DS transcripts of all severity groups ($P$ < 0.001 across all three groups) and DTEs of moderate and mild compared with the global events ($P$ = 0.012 and <0.001, respectively). The observed changes in the distribution of splicing events indicate potential alterations in transcript isoform diversity, likely influencing the functional repertoire of transcripts associated with severe disease.

Our previous investigation identified preferential enrichment of repeat elements, specifically short interspersed nuclear elements (SINEs), near the splicing sites of differentially expressed transcripts in the COVID-19 patients (27). This observation prompted us to investigate the overall presence of repeat elements, specifically within 200 bases downstream of 5′ splice sites (5′ss) and upstream of 3′ splice sites (3′ss) of DTEs across dengue severity (Fig 4A). Intriguingly, we noted a substantial increase in the overall abundance of repeat elements in significant DTEs of severe patients when compared to their non-significant isoforms (DIs) (Fig 4B). This contrasted sharply with DTEs in mild and moderate patients, where we observed significant depletion of repeat elements at both the 5′ss and 3′ss. We then examined the distribution of specific repeat classes, including simple repeats, LTRs, SINEs, and long interspersed nuclear elements (LINEs), at the 5′ss and 3′ss. In the severe group, we observed abundance of LINEs ($P$ < 0.001) and LTR elements ($P$ < 0.001), accompanied by depletion of SINEs ($P$ < 0.001) and simple repeats ($P$ = 0.038) in significant DTEs at 5′ss compared with DIs (Fig 4C). Conversely, in the moderate group, there was notable enrichment of SINEs ($P$ < 0.001), whereas both moderate and mild groups exhibited significant depletion of LINE ($P$ = 0.014 and 0.009, respectively) and LTR elements ($P$ < 0.001 in both) at the 5′ss of significant DTEs. At 3′ss, SINE depletion ($P$ < 0.001) and LTR enrichment ($P$ < 0.001) were observed in the severe patients (Fig 4D). On the contrary, in the moderate group, there were SINE enrichment ($P$ = 0.004) and LTR reduction ($P$ = 0.0016) at the 3′ss of DTEs. Significant DTEs in the patients with mild symptoms displayed LINE depletion ($P$ < 0.001) and LTR enrichment ($P$ < 0.001), along with simple repeat enrichment ($P$ = 0.035) at 3′ss. Thus, the alteration observed in the repeat distribution patterns across 5′ss and -3′ss from mild to severe patients could potentially be associated with alternative splicing events and isoform diversity in DTEs across severity groups. Understanding how these diverse transcript biotypes collectively contribute to the ultimate gene function is crucial.

## Differential isoforms impact on overall gene expression and functional outcomes

Given the substantial functional changes observed in DS transcripts across different levels of severity, we further examined the expression patterns of diverse transcript isoforms to comprehend the functional consequences of genetic heterogeneity arising from transcript diversity on dengue disease severity. To assess this, we examined the DTU across different severity levels. The DTU provides an accurate measurement of how individual transcript isoforms contribute to the overall gene expression (Fig 5A). In total, we identified 121 DTU events among the expressed transcripts in severe versus mild, 92 DTU events in moderate versus mild, and 7 DTU events in severe versus moderate patients (Fig 5B and Table S6). It is notable that there was a minimal overlap between the groups and no overlaps across all three comparisons. Interestingly, we observed varying degrees of an overlap between DGE, DTE, DS, and DTU events across paired comparisons between the three severity groups (Table S7). Specifically, we identified 40 genes that exhibited both DS and DTU events in the severe versus mild comparison, and only nine genes in moderate versus mild and no overlaps in severe versus moderate groups (Fig 5C–E). This underscores the inherent complexities/dynamics associated with gene expression, emphasising that no single regulatory mechanism at a particular stage can be deemed to exert sole dominant influence over the ultimate expression and functional identity of a gene.

Nevertheless, given the substantial diversification of functional pathways associated with DS transcripts in severe dengue (Fig 3B), we chose to explore the functional enrichment or reduction in specific splicing biotypes among genes that exhibited both DS and DTU events (Fig 5F). Remarkably, we observed a severity-dependent, significant decrease in the overall abundance of protein-coding biotypes within various functional pathways related to dengue pathogenesis and the host immune response. This suggests a potential impairment of these human host responses in severe dengue. We observed specific ablation of protein-coding DTU and DS transcripts involved in blood clotting, a process in severe dengue that could potentially influence increased haemorrhagic response in severe dengue. Despite the observed up-regulation of DTE related to the cell cycle and chromatin modifications in severe dengue (Fig 2G), transcript usage analysis revealed that DS transcripts involved in epigenetic regulation and the cell cycle exhibited higher abundance of non-functional biotypes such as protein-coding CDS not defined, and retained intron, respectively, in severe patients (Fig 5F). Our observations also indicate a general decrease in functional biotypes related to cell adhesion,

plot illustrates the number of exons in DTEs represented in yellow, as opposed to their corresponding non-significantly expressed isoforms, referred to as differential isoforms (DI), shown in green. This comparison is made across the severity groups. **(I)** Violin plot illustrates the ratio of isoforms expressed across severity groups. The yellow colour represents significantly expressed isoforms (DTEs), whereas green represents non-significantly expressed isoforms (DI), in comparison with the total number of possible isoforms. A Mann–Whitney $U$ test was used to compare for significance, and only significant associations are shown.

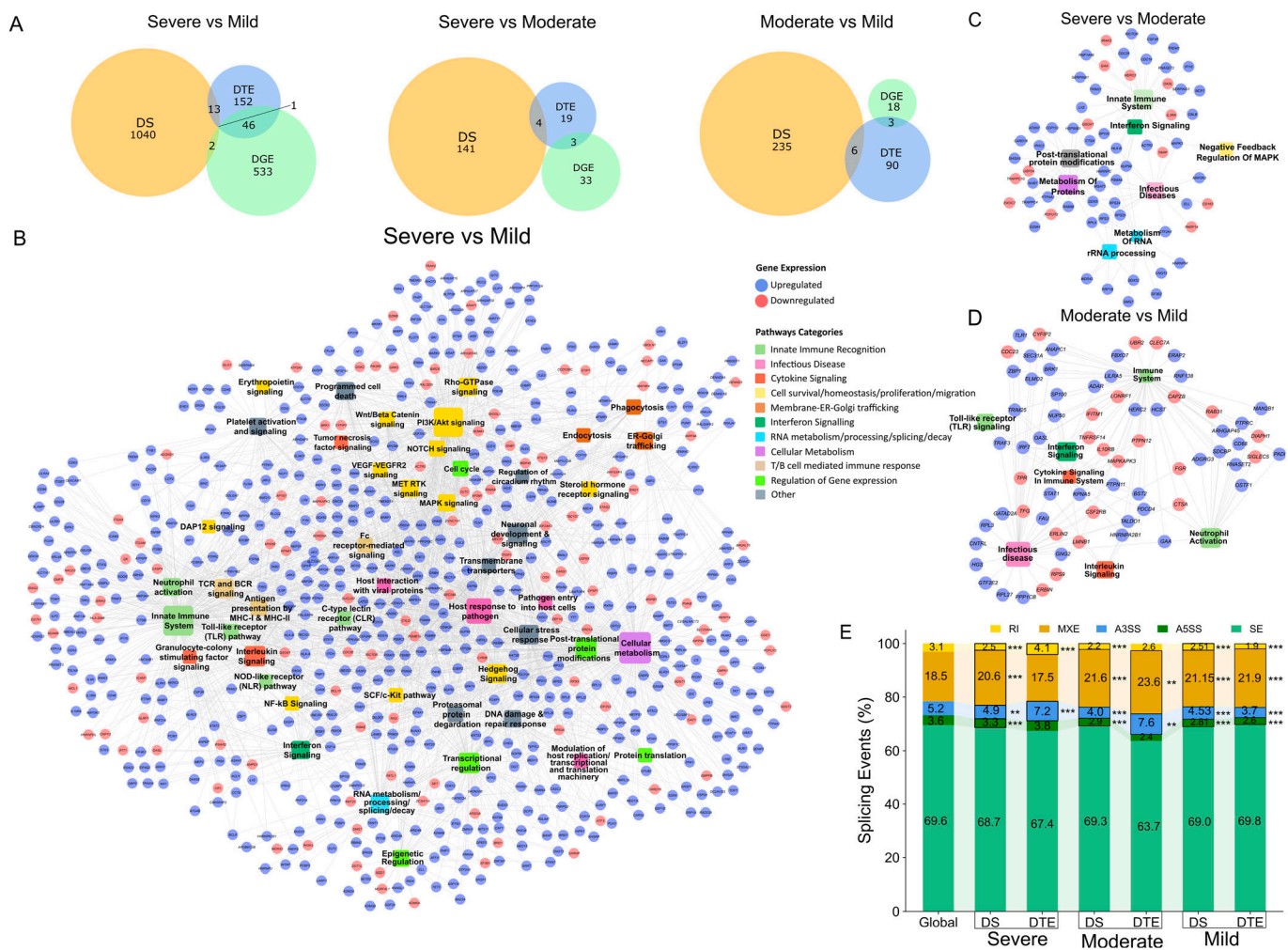

**Figure 3. Dengue severity–associated alternative splicing patterns and their functional consequences.**
**(A)** Venn diagram represents the overlap between differentially spliced (DS) genes (in yellow), differentially expressed transcripts (in blue), and differentially expressed genes (in green) across disease severity group comparisons, severe versus mild, severe versus moderate, and moderate versus mild. **(B, C, D)** Network represents the genes differentially spliced between different severity comparisons. The nodes in square represent the pathway to which the genes belong, whereas the size represents the number of genes differentially spliced in that pathway. **(B, C, D)** Node colour represents pathway categories; edges (circle) represent the gene names, and the edge colour represents direction of regulation—up-regulated (blue) or down-regulated (red)—in (B) severe versus mild, (C) severe versus moderate, and (D) moderate versus mild. **(E)** Stacked bar plot depict distribution of splicing events (retained intron [RI] [yellow], mutually exclusive exons [brown], alternative 3′ splice site [A3SS] [blue], alternative 5′ splice site [A5SS] [dark green], and skipped exons [SE] [light green] in DS and differentially expressed transcript groups compared with global transcriptome diversity across severe, moderate, and mild). The stacked bars with black borders represent the significant splicing events when compared to the global distribution of splicing events.

cytoskeletal remodelling, cellular metabolism, and intracellular signalling functions during severe dengue infection.

Furthermore, there was suppression of protein-coding biotypes associated with innate and adaptive immune responses, which could potentially drive attenuated immune responses resulting in severe infection. Surprisingly, we also observed a decrease in the protein-coding biotypes of transcripts associated with immuno-suppression in severe dengue patients. In transcripts related to RNA processing, we observed increased abundance of functional protein-coding biotypes and the elimination of non-functional NMD transcripts. This result could potentially account for the dramatic enrichment of DS genes in severe patients attributable to enhanced splicing.

## Functional consequence of DS, DTE, and DTU events on gene expression underlying dengue disease severity

To elucidate the functional implications of the DS and DTU with respect to disease severity, we focused on two genes, *MX1* and *RBM39*, which are key components of the host immune response. As both these genes also exhibit DTE between the severe and mild patients but showcasing distinct directions of regulation, *MX1* shows down-regulation, whereas *RBM39* shows up-regulation in the severe patients compared with the mild. The *MX1* gene, recognised as a type I/type III interferon–inducible viral restriction factor, demonstrates GTPase activity known to impede viral replication by disrupting the formation of the viral

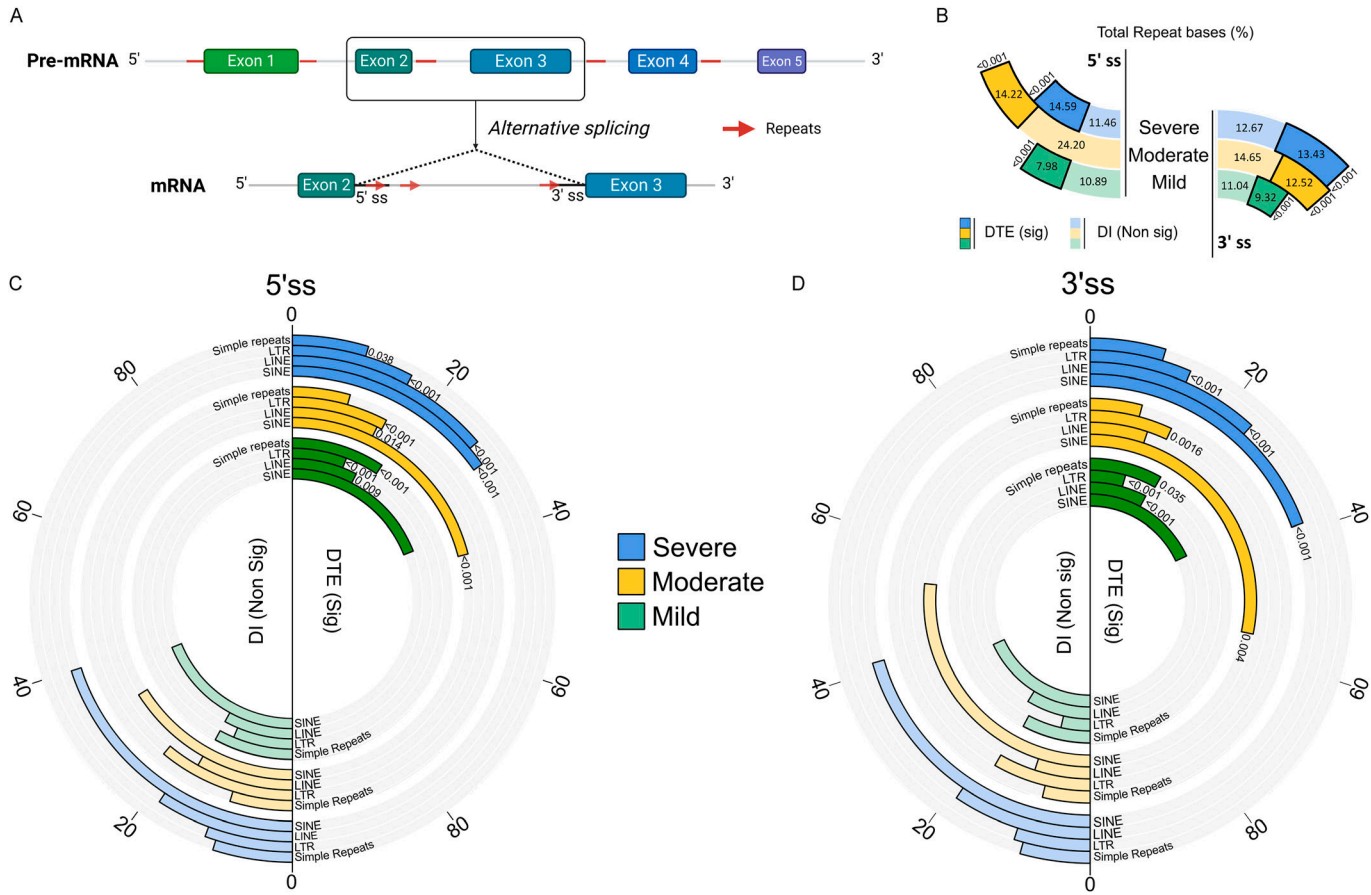

**Figure 4. Enrichment of repeat elements around the splicing sites as a modulator of alternative splicing during dengue severity.**
**(A)** Graphical illustration expands on two exonic regions to highlight the enrichment of repeat elements around the 5′ and 3′ splicing sites as a potential modulator of alternative splicing. **(B)** Stacked bar plot illustrates the proportion of total repeat bases around 200 bases downstream of the 5′ splice site (5′ss) and 200 bases upstream of the 3′ splice site (3′ss) between significant differentially expressed transcripts (with black border) and non-significant DIs (without black borders) across severity groups: severe (blue), moderate (yellow), and mild (green). **(C, D)** Radial bar plot compares the proportion of repeat classes, including short interspersed nuclear elements, long interspersed nuclear elements, LTR, and simple repeats, between significant differentially expressed transcripts (right panel) and non-significant DIs (left panel) at (C) the 5′ splice site (5′ss) and (D) the 3′ splice site (3′ss) across severity groups: severe (blue), moderate (yellow), and mild (green).

polymerase–ribonucleoprotein complex (28, 29, 30). Notably, differential intron excision was observed at two specific junctions of the gene (Fig S2A), where both the junctions corresponded to the smallest protein-coding isoform (*MX1-216*) and the significantly expressed transcript being up-regulated in the mild patients (Fig S2B). The differential splicing generates multiple transcript biotypes in the *MX1* gene, predominantly comprising protein-coding variants and one retained intron, collectively resulting in overall suppression of *MX1* gene function in severe conditions (Fig S2C).

On the contrary, the RNA binding motif protein 39 (*RBM39*) gene, identified as a novel regulator of innate immune responses and alternative splicing during viral infection, has been demonstrated to directly influence the transcription and/or splicing of interferon-stimulated genes, notably affecting the expression of interferon regulatory factor 3 (31 Preprint). *RBM39* displays distinct exon usage patterns associated with exons in the shortest protein-coding isoform (*RBM39-214*), and these patterns are also significantly differentially expressed (Fig 6A).

*RBM39* splicing activity is potentially linked to the efficient recognition of 3′ splice sites of RNA (32). It exhibits high expression in the immune compartment, including natural killer cells, T and B cells, myeloid cells, and other CD34-positive haematopoietic cells. Although an up-regulation of splicing around protein-coding transcripts is observed, differential splicing contributes to NMD biotype diversity (Fig 6B). However, a down-regulation in an NMD biotype is noted across severity, coupled with an up-regulation in protein-coding transcripts (Fig 6C). Differential splicing events in both the genes contribute not only to an increase in transcript biodiversity but also to the regulation of differential gene expression. Solely focusing on gene expression analysis might overlook the nuances associated with these transcript biotypes, which, when considered collectively, provide a more comprehensive understanding of the host response at the gene expression level. By examining results at the transcript level, we can pinpoint specific isoforms that play crucial roles in the major functions of these genes underlying differential dengue disease severity.

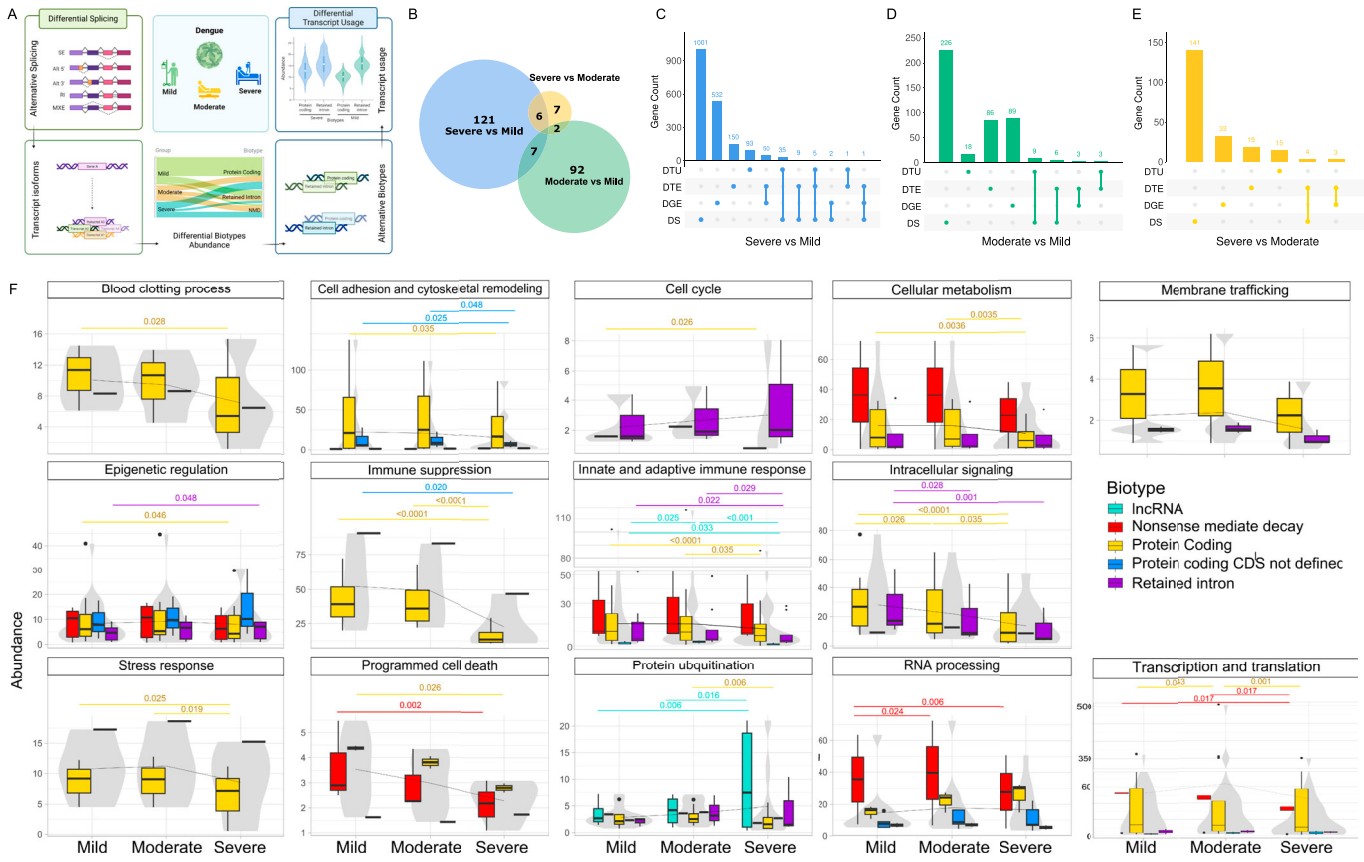

**Figure 5. Differential contribution of transcript biotypes/isoforms to the gene-level function.**
**(A)** Visual representation illustrates the role of alternative splicing in increasing isoform diversity, influencing the abundance of various transcript biotypes, and contributing differentially to overall gene expression. **(B)** Venn diagram shows the overlap between genes exhibiting differential transcript usage patterns across different severity comparison groups: severe versus mild, severe versus moderate, and moderate versus mild. **(C, D, E)** Upset plot illustrates the overlap and unique genes displaying differential transcript usage, differential transcript expression, differential gene expression, and differential splicing (DS) across the severity comparisons: (C) severe versus mild, (D) moderate versus mild, and (E) severe versus moderate. **(F)** Combined box-and-violin plot groups genes that are differentially spliced and exhibit differential transcript usage into similar functional pathways. The box represents different biotypes—lncRNA (light blue), nonsense-mediated decay (red), protein coding (yellow), protein-coding CDS not defined (blue), and retained intron (violet). The violin plot depicts the overall abundance of gene groups in severity sub-phenotypes: severe, moderate, and mild.

# Discussion

Despite being a growing global health concern over the last decade, dengue lacks specific treatments tailored to the virus (33). This becomes especially important in the background of the global COVID-19 pandemic, high population density in the tropical countries with a high incidence of dengue, and climate change–induced longer warm periods facilitating vector (mosquito) population. The absence of targeted therapies for dengue stems from a limited understanding of the mechanisms of underlying human host–DENV interactions and the associated pathogenesis during differential disease severity. Transcriptomic studies have implicated specific genes for disease severity; however, such studies have not explicitly explored the inter-individual variability leading to diverse disease severity. It is important to investigate the causes of diverse disease severity in patients who are infected with the similar/same primary pathogen. Thus, in this study, we aimed at elucidating a comprehensive transcriptomic landscape in dengue-infected hospital-admitted patients with clinical data-based severity sub-phenotype classification to uncover intricate dynamics associated with the differential expression of the transcript biotype, alternative splicing, differential transcript usage, and their potential impact on the gene functions, which ultimately leads to disease severity.

In our age- and gender-matched study cohort, we observed a correlation of increased severity with low platelet counts and leukopenia. Studies have attributed low platelet and leucocyte counts to overall reduced haematopoiesis in the bone marrow because of the dengue virus (DENV) directly infecting bone marrow progenitor cells, thus impairing their differentiation to mature blood cells (34, 35). Specifically, during the acute phase of the infection, DENV has been documented to induce bone marrow hypoplasia (36, 37). Moreover, reduced haematopoiesis has also been shown to be a part of the protective response induced by the host in response to the infection (38, 39). Ablated haematopoiesis reduces the rate of DENV infection while facilitating clearance of infection by the immune system. Interestingly, although we observed a reduction in total leucocyte counts with increased severity,

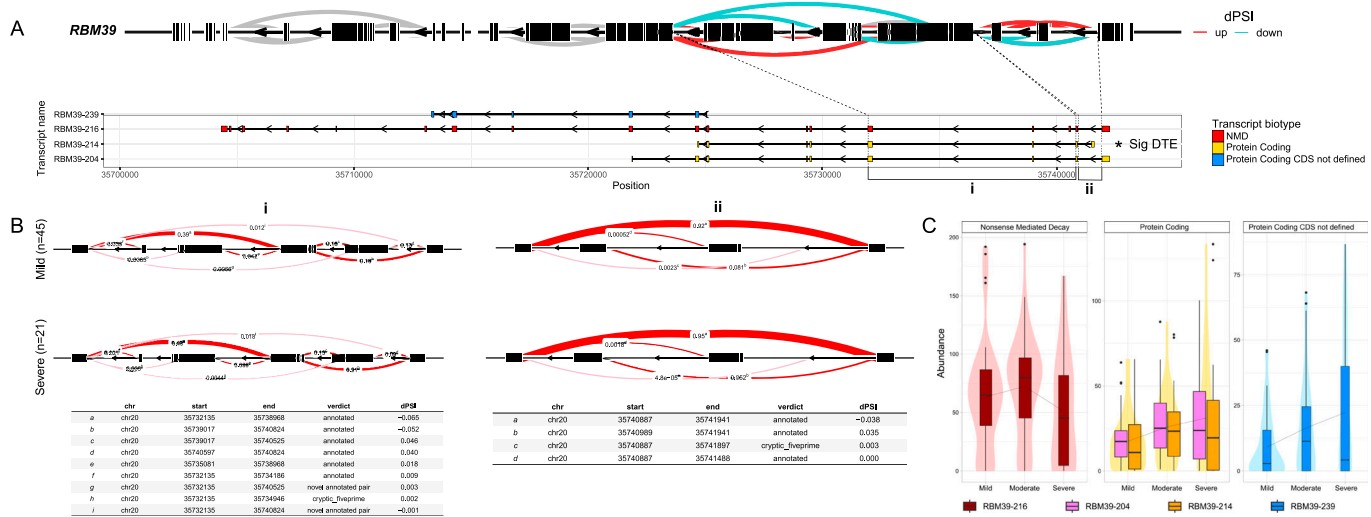

**Figure 6. Summary of the alternative splicing and differential transcript usage across dengue disease severity.**
**(A)** Sashimi graph visually underscores the *RBM39* gene, accentuating the patterns of differential intron excision in the severe versus mild group. In this representation, arcs symbolise splice junction–connected exons, and their colour indicates whether they are up- or down-regulated in the mild group. Transcripts are colour-coded based on biotypes, with nonsense-mediated decay in red, protein coding in yellow, and protein-coding CDS not defined in blue. The isoform significantly expressed is marked as *Sig differentially expressed transcript. **(B)** LeafCutter analysis depicts specific splice junctions that are differentially spliced, and the table represents the dPSI scores for each junction. **(C)** Boxplots showcase the abundance of all transcript isoforms grouped by biotypes across severity groups: mild, moderate, and severe.

there was a simultaneous increase in lymphocyte counts, indicating a rise in atypical lymphocytes. Studies have associated higher atypical lymphocyte counts with faster recovery and shorter hospital stays (40). These atypical lymphocytes have been identified as polyclonal CD19[+] B cells, and their expansion has been observed in severe DENV infections (41). The prevailing hypothesis posits that these expanded lymphocytes trigger an antibody-mediated immune reaction against the dengue virus, providing a plausible explanation for the notable increase in anti-dengue immunoglobulin G (IgG) antibodies, particularly evident in secondary dengue infections. Moreover, the expansion of CD8[+] T-cell subsets, particularly the effector memory subset, has been observed during DENV2 infection (42). The dysregulated and exacerbated immune response also potentially contributes to severe dengue pathogenesis (43). We also observed neutropenia in the severe group; DENV has been known to trigger neutrophils to release neutrophil extracellular traps, consisting of nuclear material and antimicrobial proteins, which effectively trap and eliminate invading pathogens in the vicinity (44). However, this defence mechanism comes at a cost, leading to the death of the neutrophil. Reduced haematopoiesis and increased neutrophil death can result in neutropenia in severe dengue (45). Furthermore, the increased levels of total and direct bilirubins noted in the severe groups in our cohort align with the reports associating elevated bilirubin levels with severe dengue (46, 47). Elevated bilirubin levels result from the increased lysis of RBCs, leading to the formation of free haemoglobin. The liver processes free haemoglobin to produce conjugated bilirubin, causing elevated levels in the bloodstream. Consequently, liver damage is a common complication of dengue, as indicated by elevated serum glutamic oxaloacetic transaminase, SGPT, and ALP values (47). Therefore, the clinical parameters show a clear distinction in disease manifestation between the mild, moderate, and severe patient groups.

Upon comparing the DTEs, we observed pervasive changes across the groups. When we examined the DGEs and DTEs across the severity subtypes, we noticed a minimal overlap between the two. Various studies, including our earlier investigation on SARS-CoV-2, have indicated limited concordance between the gene and transcript expressions, which may be attributed to variations in isoform-specific information during gene-level quantification, potential technical differences such as sequencing depth and sensitivity, and fluctuations in transcript expressions (48, 49). Interestingly, upon further comparing the transcript biotypes between the significant and non-significantly expressed DTEs, we observed a unique pattern of enrichment for retained intron and NMD biotypes in the severe patients, in contrast to the protein-coding biotypes in mild and moderate. This unique pattern may be attributed to altered splicing efficiency, potentially influenced by the NS5 interaction with the host U5 snRNP core splicing complex, leading to an increased rate of intron retention upon infection (15). Furthermore, recent studies have established associations between increased intron retention and pathological conditions (50).

NMD, an evolutionarily conserved pathway, identifies and eliminates transcripts with premature translation termination codons. Although initially considered an RNA quality control mechanism for removing defective transcripts, recent research has implicated this pathway in the removal of normal transcripts with specific features, such as long 3′ UTRs and GC-rich regions (51). Importantly, the host strategically employs the NMD pathway as a defence mechanism upon interaction with the positive-strand RNA viruses (52). Up-regulation of the host NMD pathway facilitates the degradation of viral transcripts, resulting in a substantial impairment of viral replication. Therefore, the down-regulation of this pathway can contribute to a higher dengue viral load in severe patients (53). This implies that dengue severity is closely linked to enrichment of

transcripts with non-functional biotypes such as NMD and retained intron, accompanied by a simultaneous reduction in protein-coding transcripts. The lower number of exons and shorter CDS length in the severe group align with our earlier findings in severe COVID-19 patients, indicating consistent transcript length patterns ([27]). However, unlike our previous observations, severe dengue patients showed increased transcript diversity, characterised by a rise in non-canonical transcripts and a decrease in protein-coding isoforms. This suggests pathogen-specific dynamics in the human host transcriptome, warranting further investigation. Although we observed significant enrichment of various immune response pathways at both the gene and transcript levels, a notable finding is the up-regulation of PCD and haemolytic response, coupled with impaired interferon and antiviral response, particularly at the transcript level. These variations in functional pathways align with the clinical symptoms presented by the patients. The up-regulation of PCD might provide an explanation for leukopenia, neutropenia, and thrombocytopenia in the severe cases (Fig 1E) ([54]). Increased haemolysis contributes to platelet aggregation, resulting in an overall reduction in platelet counts in the blood (Fig 1C) ([55]). The abnormal lymphocytosis could be linked to an exacerbated immune response, including B-cell signalling (Fig 2G), leading to its atypical expansion, as discussed earlier. These regulations could be driven by the invading virus to facilitate rapid synthesis of viral transcripts, contributing to the dissemination of the virus within the host and leading to severe disease pathology. Nevertheless, it is essential to consider that alterations in the host transcriptome landscape may arise as a consequence of the host's response to combat severe infection, necessitating the swift activation of multiple defence pathways, including the immune response. These findings shed light on the complex interplay between the host immune response and viral modulation of the splicing machinery, providing insight into the pathogenesis of severe dengue.

However, upon comparing splicing patterns in the transcripts, we observed a minimal overlap between the DS and DE genes/isoforms. This is possibly because splicing and expression are presumably independently regulated, which may have stronger functional impacts on the disease severity. Although it is anticipated that viruses manipulate the host's spliceosomal machinery to enhance their survival and replication, we also detect significant splicing alterations in the genes related to the host's immune regulatory functions. Delineation of the splicing events influenced by DENV for its advantage compared with those that contribute to the host-driven defensive response aimed at clearing the virus is crucial for unravelling the mechanisms contributing to the development of severe states in DENV infection ([14], [56]). Specifically, intron retention is harnessed to play regulatory roles in the innate immune response, for example, in macrophage functioning, and decreased IR in nuclear-detained mRNA has been coupled with the increased expression of phagocytosis and inflammatory signalling genes ([57]); however, as previously mentioned, viral NS5 protein can counter this by interacting with the core host splicing component. The notable enrichment of A3SS events in genes expressed during increased severity raises questions about their role in host immune responses or viral pathogenesis, highlighting the need for further investigation. The potential role of *RBM39* in this context during dengue infection warrants exploration. Existing studies have suggested that the higher GC content near the splice site is associated with the regulation of alternative splicing, possibly through the formation of secondary structures influencing relative spatial positioning of genes and splicing choices ([58], [59], [60]). Notably, we observed enrichment of GC-rich repeat elements, such as LINEs and LTR elements around both 3' and 5' splice sites in severe DTEs compared with non-significant DIs, prompting a deeper exploration of their influence on transcript diversity during disease severity.

As a specific instance of alternative splicing, the transcript usage pattern across severity reiterated critical observations from earlier analyses. We observed a significant severity-dependent decrease in the overall abundance of protein-coding biotypes within the functional pathways associated with both dengue pathogenesis and the host immune response. This observation implies a potential impairment of these critical responses in cases of severe dengue. It is worth noting that in a previous analysis, we identified an increase in the expression of genes associated with immune response pathways in severe dengue patients (Fig 2G). However, measuring gene expression levels alone doesn't precisely reveal the functional biotype of differentially expressed transcripts. In our samples, severe dengue was characterised by a reduction in the abundance of functional protein-coding transcript biotypes, suggesting potential impairments in the functions associated with these transcripts (Fig 2F). This observation implies an ongoing active immune response, which contradicts the existing state of severe disease in these patients. It is known that the pathogenesis of many viral infections, including dengue, is not primarily driven by a viral load. Rather, dysregulated immune response and consequent exacerbated inflammation lead to widespread tissue damage. Therefore, the depletion of protein-coding biotypes involved in immunosuppression in severe dengue could potentially result in an uncontrolled immune response and consequent severe clinical pathology.

A study conducted by Kanlaya et al revealed co-localisation between the DENV capsid and Mx1/MxA protein in human endothelial cells, suggesting potential inhibition of Mx1 function by the virus ([61]). Reduced *MX1* expression may potentially result from a diminished antiviral interferon response, contributing to the severe phenotype of dengue. Although *RBM39* has been demonstrated to regulate cell cycle and DNA damage response pathways ([62]), a recent study by Song et al indicated that the increased expression of *RBM39* down-regulated the expression of proinflammatory cytokines and enhanced viral replication in porcine reproductive and respiratory syndrome virus (PRRSV) infection ([63]). Therefore, it is essential to conduct additional investigations into the involvement of *RBM39* in the innate immune response, which contributes to viral pathogenesis.

Although the study's cross-sectional design provides valuable insights into the transcriptomic changes occurring during dengue infection, it has potential limitations in capturing the dynamic evolution of these alterations over time, elucidating cell type–specific variations in splicing and expression. To strengthen the results, independent studies with their requisite in-build validations are essential, especially considering the use of best possible, yet different, algorithms for gene and transcript expression analyses. In the absence of the same/similar tool for both the

investigations, distinct computational methods may introduce variability and potential biases into the results. Validating these findings through independent experiments or employing alternative analytical approaches would enhance the robustness of the findings. A crucial consideration for future research could involve adopting a cell type–specific temporal study design. Thus, to gain a more holistic understanding of the temporal dynamic, longitudinal patient profiling will be more insightful. Furthermore, longitudinal single-cell–based studies will provide supplementary insights into the inferences highlighted in this study. In addition, the absence of transcript-specific functional pathway databases poses a challenge in accurately interpreting the biological functions of identified transcripts. Addressing this limitation requires concerted efforts to develop or enhance databases tailored to transcriptomic data. Furthermore, the study emphasises the importance of exploring pathogen-specific dynamics alongside the host response to gain additional insights into dengue pathogenesis. Investigating the intricate interplay between host–virus interactions could reveal novel mechanisms contributing to disease severity.

In summary, our study provides a comprehensive insight into the transcriptomic landscape of dengue-infected severity subphenotypes, revealing intricate dynamics that influence disease severity. The patterns observed in the differential expression of transcripts, alternative splicing, and transcript usage contribute to the complex interplay between the host immune response and viral manipulation, offering valuable insights for potential therapeutic targets in severe dengue cases.

# Materials and Methods

### Sample collection and clinical classification

The present investigation, carried out at CSIR-Institute of Genomics and Integrative Biology (IGIB) in New Delhi, India, included a cohort of 112 hospital-admitted individuals diagnosed with dengue through NS1 antigen testing. After the acquisition of written consent from the participants, the paramedical team at MAX Super Specialty Healthcare Hospital, New Delhi, India, gathered 2–3 ml blood samples during the initial hospital visit using EDTA-coated vials, adhering to the principles outlined in the Declaration of Helsinki. This sample collection took place between August 2022 and November 2022, which coincides with the peak dengue infections in New Delhi, India. All pertinent clinical information was retrospectively obtained by scrutinising the electronic medical records of each participant.

Based on the complete blood count reports of the patients, the study categorised the 112 dengue NS1-positive patients into three groups: *dengue without warning signs* (45 patients with normal platelet and leucocyte counts), and *dengue with warning signs* (comprising two sub-groups—46 patients with normal platelet but decreased leucocyte counts [leukopenia] and 21 patients with both low platelet counts [thrombocytopenia] and decreased leucocyte counts [leukopenia]). None of the patients exhibited severe bleeding, organ failure, or abnormal liver parameters, in accordance with the World Health Organization (WHO) dengue

classification and management scheme (5). Consequently, a separate group for severe dengue was not formed. For clarity, the groups are designated as mild (no thrombocytopenia and leukopenia), moderate (no thrombocytopenia but leukopenia), and severe (thrombocytopenia and leukopenia) dengue cases.

### Nucleic acid isolation

RNA extraction from the blood samples was carried out using QIAGEN QIAamp RNA Blood Mini Kit, following the manufacturer's instructions with specific modifications. The incubation time and centrifugation time during the erythrocyte lysis step were reduced to 5 min. In addition, for enhanced RNA purification, a 3-min incubation period was introduced during all washing steps. The purity of the isolated RNA was assessed using NanoDrop Microvolume Spectrophotometer and confirmed by running it through an agarose gel. Subsequently, the RNA was stored for short time at −80°C until thawed for total RNA-seq library preparation.

### Library preparation and sequencing

Libraries were prepared with a total of 250 ng of total RNA using Illumina TruSeq Stranded Total RNA Library Prep Globin Kit (Cat. No. 20020612; Illumina). Initial steps involved the depletion of globin mRNA and ribosomal RNA (both cytoplasmic and mitochondrial), as these two forms of abundant RNA are present in high levels in the whole blood. Cleaved RNA fragments underwent first-strand cDNA synthesis through reverse transcriptase and random primers. Subsequently, double-stranded cDNA was synthesised using DNA polymerase 1 and RNase H, followed by purification using AMPure XP (A63881; Beckman Coulter). Adenylation at the 3′ blunt end of double-stranded cDNA was performed, and sequencing libraries were uniquely indexed and enriched through PCR amplification.

Library quality was confirmed through size analysis on Agilent 2100 Bioanalyzer, and library concentrations were determined using the Qubit double-stranded DNA (dsDNA) high-sensitivity (HS) assay kit (catalogue no. Q32854; Thermo Fisher Scientific). Libraries were then diluted to 4 nM and combined in an equimolar fashion, with 24 samples per library pool. Paired-end, 2 × 151 read length sequencing was performed on a NextSeq 2000 platform (Illumina) at a final loading concentration of 650 pM.

### Differential gene and transcript expression analyses

The raw sequencing reads were demultiplexed using bcl2fastq (v), and subsequently, read quality was assessed using FastQC (64, 65). To filter out low-quality bases (phred score < 20) and trim any adapter content, Trimmomatic v.0.39 was used (66). The filtered reads underwent a quality check to ensure effective filtration, and quantification was carried out using the pseudo-alignment method in Salmon (v1.8.0) against the human reference genome (GRCh38.110.primary assembly from Ensembl) (Fig S3) (67).

During quantification with Salmon, the −numGibbsSamples parameter was set to 10 inferential replicates to generate bootstrapped abundance estimates for each sample using posterior Gibbs sampling to account for the variability in transcript abundance. The quantified files were imported to the R environment

using the tximport R package for differential expression (DE) analysis (68).

For gene-level DE analysis, the DESeq2 package was employed and genes with >10 reads in at least 21 samples (the size of the smallest group) were retained for stringent further analysis (69). To control batch effects in the data, the removeBatchEffect function from the limma R package was used in conjunction with the model design in DESeq2 (70). For transcript-level DE analysis, the swish method from the fishpond (v 1.0.0) R package was used, using the inferential replicates (71). Similar to DESeq2 analysis, transcripts with >10 reads in at least 21 samples were kept for differential expression analysis. Unannotated significant genes were further filtered.

Differential gene and transcript expression analyses were conducted between different severity patients' groups, namely, severe versus mild, severe versus moderate, and moderate versus mild. The Benjamini–Hochberg (BH) correction was applied to address multiple comparisons. DEGs and DETs with an adjusted $P <$ 0.05 and $| \log_2$ fold change $(\log_2 FC)| \geq 1.5$ were considered statistically significant.

## Gene set enrichment analysis

Gene set enrichment analysis using the filtered lists of differentially expressed (DE) genes and transcripts was performed. The human MsigDB collection, specifically the Reactome pathway database (c2.cp.reactome.v2023.1.Hs.symbols.gmt), was employed for this analysis through the fgsea R package (72, 73 Preprint). DE statistics served as the basis for ranking genes, and for transcripts, the average of only the significant isoforms belonging to the same gene was considered for ranking. Pathways with at least two genes mapping to the pathway and a $P$-value cut-off of < 0.05 were considered significant (Table S4). Redundant or similar pathways were grouped together for clarity. We used ggplot2 in R to visually represent the significant pathways, considering both the combined score and the number of genes involved in these pathways (74).

## Differential splicing analysis

The filtered fastq files were aligned against the reference human genome (GRCh38.110) using the splice-aware aligner STAR (v2.7.11) (75). The aligned bam files were used for detecting both event-based splicing detection and differential splicing quantification between the severity groups. For event-based differential splicing detection, rMATS (v4.2.0) (replicate Multivariate Analysis of Transcript Splicing) was employed with the --variable-read-length and --allow-clipping parameters (76). The statistical model of rMATS detects differential splicing by employing a hierarchical framework to model exon inclusion levels, considering estimation uncertainty and variability among replicates.

For quantification of differential splicing between severity groups, the LeafCutter v0.2.9 tool was used (77). For this, the bam files were transformed into junction files using regtools. Subsequently, intron clustering and analysis of differential splicing were performed using LeafCutter, with allowed intron sizes set between 50 and 500 kb. The human reference was used for annotation. The groups for differential splicing analysis were organised in a manner

consistent with those used for the comparison in the differential gene/transcript-level analysis. The results were visualised using LeafViz, and a report was generated using RStudio.

## Viral read quantification

Using the reads unmapped to humans, alignment was performed against the serotype-specific DENV genomes for DENV1, DENV2, DENV3, and DENV4 (NC_001477, NC_001474, NC_001475, NC_002640) using BWA-MEM to generate bam files (78). The viral genome coverage and depth were determined using SAMtools (Table S1) (79).

## Repeat element enrichment analysis

We analysed the repeat elements around the splicing sites for the DTEs and their non-significantly expressed isoforms (referred to as differential isoforms [DIs]). The genomic sequences 200 bases upstream of the 3′ splice site and 200 bases downstream of the 5′ splice site were extracted for each transcript from the GRCh38 reference genome using BEDTools (80). To query for repeat elements, all the sequences were uploaded to RepeatMasker using hmmer search engine and Human as DNA source (81). The repeat elements considered were LINEs (L1 and L2), SINEs (Alu and MIR), LTR, and the simple repeats (Table S8). The repeat bases were normalised for total repeat bases and represented as proportion. The significance was compared between DTEs and the DIs using the $\chi 2$ test in GraphPad Prism, and the data were visualised in Excel and modified using Inkscape.

## Differential transcript usage analysis

Transcript usage was evaluated, and variations in the proportional impact of a transcript on the overall gene expression were computed by determining its relative abundance. The DTU analysis was performed using the salmon alignment-free abundance estimations, employing direct read quantification at the transcript level. The DRIMSeq R package was used for the analysis, relying on a Dirichlet-multinomial model for individual genes and estimating a gene-wise precision parameter (82). This approach directly modelled the correlation among transcripts within their parent gene.

To mitigate false discovery rates stemming from the complexity of transcript diversity and the number of transcripts per gene, transcripts with fewer than 10 read counts or with a relative contribution to the overall gene expression of less than 0.1 were excluded. In addition, genes with fewer than 10 counts in any samples were omitted. The resulting filtered set of transcript-level counts was used as input for the DRIMSeq analysis to investigate alterations in transcript usage between severity groups. The P-values reported at the gene level were further corrected within stageR using the BH FDR procedure (83). For the screened genes, transcript-level $P$-value is adjusted to control for a family-wise error rate. A transcript was deemed a DTU event if the $P$-value was less than 0.05. A gene was considered differentially used if it had at least one DTU event.

## Pathway enrichment analysis

Pathway enrichment analysis for all the differentially expressed genes (DS genes) was performed using the Enrichr web tool with the Reactome database (84). Pathways with a *P*-value cut-off < 0.05 were considered significant. To avoid redundancy, similar biologically functional pathways were grouped together (Table S9). The pathways were visualised as a network using Cytoscape (v3.10.1) (85). Furthermore, pathway enrichment for DTU events that overlapped with DS was conducted in a similar manner. Gene functions for all transcripts were depicted using the ggplot2 R package. Significance between transcripts and severity groups was assessed using the non-parametric two-way ANOVA test with Tukey's multiple test correction.

## Statistical analysis and data visualisation

Wherever appropriate, differences between continuous data points were assessed using the two-tailed Mann–Whitney *U* test, whereas categorical data comparisons were conducted through chi-square testing. All statistical analyses were carried out using a licensed version of GraphPad Prism. For data visualisations, a combination of ggbio (86), GenomicRanges (87), biovizBase (88), ggplot2 (74), ggbreak (89), ggraph (90), dendextend (91), igraph (92), colormap (93), EnhancedVolcano, GViz (94), rawgraphs (95), SRPlots (96), and Microsoft Excel was employed. Figures were subsequently modified using Inkscape (97). A significance threshold of *P* < 0.05 was applied, unless otherwise specified.

# Data Availability

The datasets presented in this study can be found online at the NCBI SRA under the BioProject accession number PRJNA1071729.

## Ethics statement

The study was designed in accordance with the Declaration of Helsinki and was approved by the institutional ethics committee of CSIR-Institute of Genomics and Integrative Biology, Delhi, India (Ref No: CSIR-IGIB/IHEC/2020-21/01). The patients/participants provided their written informed consent before participation in this study.

# Supplementary Information

# Acknowledgements

The authors duly acknowledge all the dengue patients who participated in the study. The authors further recognise Aanchal Yadav, Priti Devi, and Aparna Swaminathan for their contributions to sample isolation, library preparation, and sequencing data collection in this study. Authors acknowledge the help and support from Dr Bharti Kumari towards facilitation as a research manager and coordination with the funders. The authors acknowledge the support of Anil Kumar and Nisha Rawat towards dengue sample transport and sample management. P Chattopadhyay acknowledges the CSIR for his research fellowship. This research was funded by Bill and Melinda Gates Foundation, grant number—INV-033578; and Rockefeller Foundation, grant number—2021 HTH 018, to R Pandey.

## Author Contributions

P Mehta: data curation, formal analysis, investigation, visualisation, methodology, and writing—original draft.
CSC Liu: investigation, literature review, and writing—original draft.
S Sinha: investigation, literature review, and writing—original draft.
R Mohite: investigation and writing—original draft.
S Arora: formal analysis and visualisation.
P Chattopadhyay: investigation and writing—original draft.
S Budhiraja: resources and writing—review and editing.
B Tarai: resources and writing—review and editing.
R Pandey: conceptualisation, supervision, funding acquisition, project administration, and writing—review and editing.

## Conflict of Interest Statement

The authors declare that they have no conflict of interest.

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
