## [Reviewer comments · Life Science Alliance]

Life Science Alliance

Reduced Protein-Coding Transcript Diversity in Severe Dengue emphasize role of Alternative Splicing

Priyanka Mehta, Chinky Liu, Sristi Sinha, Ramakant Mohite, Smriti Arora, Partha Chattopadhyay, Sandeep Budhiraja, Bansidhar Tarai, and Rajesh Pandey

DOI: <https://doi.org/10.26508/lsa.202402683>

Corresponding author(s): *Rajesh Pandey, Institute of Genomics and Integrative Biology*

Review Timeline:

Submission Date:	2024-02-28
Editorial Decision:	2024-05-06
Revision Received:	2024-05-16
Editorial Decision:	2024-05-21
Revision Received:	2024-05-22
Accepted:	2024-05-22

Scientific Editor: *Eric Sawey, PhD*

Transaction Report:

May 6, 2024

Re: Life Science Alliance manuscript #LSA-2024-02683-T

Dr. Rajesh Pandey
Institute of Genomics and Integrative Biology
Mall Road, Delhi
Delhi 110007
India

Dear Dr. Pandey,

Thank you for submitting your manuscript entitled "Diminished Protein Coding Transcript Diversity in Severe Dengue Patients highlight functional role of Alternative Splicing" to Life Science Alliance. The manuscript was assessed by expert reviewers, whose comments are appended to this letter. We invite you to submit a revised manuscript addressing the Reviewer comments.

Thank you for this interesting contribution to Life Science Alliance. We are looking forward to receiving your revised manuscript.

Sincerely,

B. MANUSCRIPT ORGANIZATION AND FORMATTING:

Reviewer #1 (Comments to the Authors (Required)):

This is a very interesting study and is quite novel and relevant given that samples from human patients infected with dengue virus are used. It is also a significant study in that it provides insight into virus tropism in the human body, which is of importance for the development of therapies that could be useful for dengue treatment, as of currently there are none. This manuscript is well-written; however, I do have some major revision requests for this manuscript before manuscript publication, please see below:

Major comments:

1. Do we know which DENV serotype was present? Only says NS1 antigen detection, not sero-specific?
2. Do we know if any of these individuals have had DENV before? Were they tested for any other viruses or on any medications (i.e. anticoagulants, steroids, antihistamines, etc.)? All of these factors also need to be taken into consideration.
3. How come there were no proper controls with no infection? It wouldn't be too much effort to collect blood from uninfected individuals.
4. Swish R was used for the differential transcript -level analysis and DESeq2 used for the differential gene-level analysis, do these programs use different algorithms (they likely do, it looks like swish is nonparametric rank-based and the normalisation that these programs perform is likely also different) and how would this affect the results? Why was Swish R not used to determine differentially expressed genes and transcripts? It makes more sense to be using the same program to make these comparisons. I also see this same problem done with the aligners, STAR is used for the alternative splicing analysis, but Salmon used for the differential transcript-level analysis. When so many different programs with different algorithms are mixed, you don't know if true differences in your results are real or by-products of using different software. Please explain how this is accounted for or detail this limitation in the discussion.
5. I realise this is a bioinformatics paper, but I would really encourage the validation of alternative splicing, differential expression, etc. with RT-qPCR. It would be best to make clear the limitation that this work is entirely based on sequencing and would need other experimental studies as follow-up.
6. Results section is a bit long (some of the information could be moved to the discussion).
7. Figure resolution will need to be improved for almost all figures, please see specific figure comments below.

Minor comments:

Line 38: "minimal overlap between the gene and transcript expressions" needs to be clarified a bit. This could imply that single nuclei and single cell sequencing was done. Also, please see my comment above about using different programs for this analysis, I recommend it was done only using Swish R.

Line 142: "gene and transcript-level differential expression analysis", once again this should be clarified, as I also don't see where this distinction is made in the Figure 1 study design overview. It would be good to get this clarity how these analyses were separated in this figure, maybe add the software/programs used into Figure 1.

Table 1 can be moved to supplemental.

Figure 2 resolution needs to be improved, especially panels A-C as the genes are labelled but they aren't readable with the size of each plot. Panel E text is very difficult to read, also is panel E necessary?

Line 234: "significant positive enrichment" p-value? This paragraph goes on to discuss all enriched pathways, how was this

enrichment defined? By how many genes in a pathway or fold-changes? If based on GSEA, there should be associated p-values.

Figure 3. Please improve the resolution of this figure. All text should be readable, this includes the numbers on panels B and F. Is panel A necessary, it seems like this just shows what alternative splicing is and doesn't add to the results?

Line 295-299: I don't think defining alternative splicing in the results is appropriate, some detail could be given in the introduction, but not necessary here.

Figure 4: Resolution of this figure could also be improved. It would be worth thinking about splitting these larger figures into different figures if the panels together are so small it creates difficulties reading them.

Figure 5. Resolution of this figure needs to be improved; panel A could possibly be removed to increase the size of the other panels to improve readability.

Figure 6. The readability/resolution of this figure needs to be improved.

Figure 7. This figure should be supplemental.

Reviewer #2 (Comments to the Authors (Required)):

The authors collected blood samples from 112 NS1 antigen positive hospital admitted Dengue patients and classified them into three groups: Dengue without warning sign referred to as Mild (n=45), Dengue with warning sign which was further divided into two groups: Moderate (n=46) and Severe (n=21). The authors employed bulk RNA-seq analysis of whole blood samples to examine transcriptomic patterns associated with the disease severity. In particular, the authors focused on the analysis of alternative splicing and used multiple published bioinformatics approaches.

However, there is a lack of evidence indicating that the isoform level analysis using the obtained bulk RNA-seq data was reliable. Further, the isoform level analysis could be complicated by using total RNAs, as it would capture many immature transcripts as well. It was unclear how well the blood samples were stored either. Rigorous validations should be provided. Also, it is unclear if some observed changes in the isoform usage were simply due to the changes in cell types in the blood samples.

Dear Editor and the Reviewers,

We take this opportunity to thank you for your time and effort towards critical review of our manuscript and important suggestions towards improving the inferences highlighted in the manuscript. We have made an effort towards addressing all the suggestions as appropriate with modified figures, supplementary, text and references.

Best wishes,

Rajesh

Reviewer #1 (Comments to the Authors (Required)):

This is a very interesting study and is quite novel and relevant given that samples from human patients infected with dengue virus are used. It is also a significant study in that it provides insight into virus tropism in the human body, which is of importance for the development of therapies that could be useful for dengue treatment, as of currently there are none. This manuscript is well-written; however, I do have some major revision requests for this manuscript before manuscript publication, please see below:

We sincerely appreciate the comprehensive review of our manuscript. Your acknowledgment of the novelty and relevance of our study, particularly in its utilization of hospital admitted human samples infected with dengue virus, is immensely gratifying. The recognition of our research's significance in elucidating virus tropism within the human body underscores its potential impact on the development of therapeutics for dengue, an area currently lacking effective treatments. We value your critical feedback and specific revision requests, as they will undoubtedly contribute to refining the clarity and rigor of our work. We have provided a detailed response to all the suggestions below with combination of additional analysis, discussion points, figure updates and clarifications.

Major comments:

1. Do we know which DENV serotype was present? Only says NS1 antigen detection, not sero-specific?

We thank the reviewer for the comment. Although Dengue serotype testing was not conducted for the patients in the hospital from where the clinical samples were collected, but based on your suggestion, we used the Dengue viral reads captured as part of our total RNA-seq data (fraction of the non-human aligned reads from the data). Analysis revealed that majorly the patients were infected with DENV2 serotype with differential genomic coverage of the dengue virus, except for some which we did not have adequate sequencing reads for the dengue serotype analysis. This is in consonance with the prevalence of DENV2 during the time of sample collection in Delhi, India.

Additionally, the viral read counts with genomic coverage and sequencing depth for each patient have been included in **Supplementary File 1**, during revised submission. But yes, this is secondary derivative about the dengue genome analysis based serotype information.

2. Do we know if any of these individuals have had DENV before? Were they tested for any other viruses or on any medications (i.e. anticoagulants, steroids, antihistamines, etc.)? All of these factors also need to be taken into consideration.

We value the reviewer's suggestion and have duly noted it. It's important to clarify that all patients enrolled in our study experienced primary infection with DENV, and this information has been explicitly stated in the results section. Regarding the absence of testing for other viruses, it's worth mentioning that the samples were collected between August and November of 2022, a period during which Dengue infection is prevalent in the Indian subcontinent. Therefore, given the context, testing for other viruses was not conducted. But, it is good idea for future studies from the lab in this direction.

Furthermore, as our study is based on a single time point of patient blood collection during first reporting to hospital with symptoms and followed by testing, information regarding administered

medications for the alleviation of dengue symptoms was not utilized to infer results. We confirmed with the clinical partner that for the patients enrolled in this study, when they arrived at the hospital, were not under medication for something else. However, we acknowledge the importance of considering these factors in future longitudinal studies which are being undertaken in the lab presently, and they will be taken into careful consideration.

3. How come there were no proper controls with no infection? It wouldn't be too much effort to collect blood from uninfected individuals.

We extend our gratitude to the reviewer for their insightful comment.

We take this opportunity to share that the study theme in our lab is to understand differential disease severity modulators albeit infected by the same/similar pathogen. The present study is in the same line for elucidating differential dengue severity albeit infected by same DENV2. That's the primary reason that the study design does not include healthy/uninfected individuals as control to align with the primary objective of our study. While the inclusion of a control group is typically valuable for distinguishing between effects attributable to the pathogen and those stemming from other factors such as host genetics or environmental influences, our study focuses specifically on elucidating transcript-level variations across different disease severities within the infected population.

While studies have examined dengue pathogenesis by comparing infected individuals to healthy controls, our study adopts a distinct approach. Our aim is to comprehensively investigate the transcript-level dynamics associated with various degrees of dengue infection severity. By exclusively analyzing individuals exhibiting different levels of disease severity, we aim to discern any differences in gene expression profiles linked to these clinical phenotypes. Our study design thus enables a targeted analysis of transcriptional dynamics directly associated with disease severity, contributing to a deeper understanding of dengue pathogenesis.

4. Swish R was used for the differential transcript -level analysis and DESeq2 used for the differential gene-level analysis, do these programs use different algorithms (they likely do, it

looks like swish is nonparametric rank-based and the normalisation that these programs perform is likely also different) and how would this affect the results? Why was Swish R not used to determine differentially expressed genes and transcripts? It makes more sense to be using the same program to make these comparisons. I also see this same problem done with the aligners, STAR is used for the alternative splicing analysis, but Salmon used for the differential transcript-level analysis. When so many different programs with different algorithms are mixed, you don't know if true differences in your results are real or by-products of using different software. Please explain how this is accounted for or detail this limitation in the discussion.

We acknowledge the reviewer's perspective and acknowledge the suggestion.

As you know, in RNA-seq analysis, addressing uncertainty associated with transcript abundance estimation is crucial. Inferential replicates are employed to capture and manage this variability, particularly in scenarios where fragments may map to multiple transcripts or genes.

To obtain these inferential replicate estimates, Salmon was used for quantification at transcript-level as it is able to provide bootstrapped/Gibbs sampling estimates which are used for differential expression at the transcript-level. Subsequently, Swish method was used as it incorporates inferential replicate counts to account for uncertainty in the abundance estimates. It extends the existing SAMseq method and has been shown to improve control of the false discovery rate, particularly for transcripts with high inferential uncertainty. This approach is particularly crucial for transcripts with multiple isoforms, high sequence homology, or those belonging to gene families, where the quantification uncertainty can vary significantly across transcripts. While gene-level analysis is less ambiguous compared to transcript-level quantification, as it avoids the complexities associated with isoform quantification.

At the gene level, the counts from all transcripts belonging to a gene are simply summed up, providing a more robust estimate of gene expression, which is better handled by the DESeq2 package. DESeq2 is known for its robust control of FDR at the gene level, ensuring reliable identification of differentially expressed genes. The nonparametric approach of swish is more suitable for transcript level analysis to capture the complexity and variability of transcript-level

expression patterns, especially in scenarios where assumptions about data distribution may not hold. The negative binomial distribution used by DESeq2 is well-suited for modeling gene expression levels, especially when the mean is less than the variance, as is often the case with RNA-seq count data.

To account for different algorithms, a similar model design was used to estimate distribution and stringent cutoffs of read counts and significance cut offs on log₂FC was applied to take only the most significantly differentially expressed genes and transcripts into consideration. However, we understand that the differences in algorithm may have some effects and hence we have also included it as a potential limitation of the study, in the absence of a tool which combines the best for both analyses.

Regarding alignment methods, STAR was employed for differential splicing analysis because of its capability as a splice-aware aligner, which is essential for aligning RNA-seq reads covering splice junctions. Differential splicing tools such as Leafcutter and rMATS require BAM files to count junction-spanning reads for differential splicing analysis. Therefore, two different alignment methods were used to accommodate the distinct requirements of transcript-level quantification and differential splicing analysis. While such alignment-based methods do not take into account inferential replicates and so for transcript-level analysis, Salmon's alignment-free methods was employed for more accurate and faster quantification.

5. I realise this is a bioinformatics paper, but I would really encourage the validation of alternative splicing, differential expression, etc. with RT-qPCR. It would be best to make clear the limitation that this work is entirely based on sequencing and would need other experimental studies as follow-up.

We appreciate the reviewer's opinion. Take this opportunity to share that yes, it is a bioinformatics based findings but all the samples collection (along with our clinical partner) happened in the lab, along with RNA-seq of those clinical samples followed by analysis.

We have included the following paragraph in the limitations section.

“To strengthen the results, independent studies with its requisite in-built validations are essential, especially considering the use of best possible, yet different algorithms for gene and transcript expression analysis. In the absence of same/similar tool for both the investigations, distinct computational methods may introduce variability and potential biases into the results. Validating these findings through independent experiments or employing alternative analytical approaches would enhance the robustness of the findings.”

As these study samples were collected in mid-2022, we do not have RNA left for the RT-qPCR validation. Yet, we have tried to introduce bioinformatics stringency during the analysis to strengthen robustness of the data and the inferences therein.

Few highlights include:

- 1) A stringent quality control protocol was meticulously applied throughout the analysis pipeline. Reads with a Phred score below 30 were filtered out, and trimming was performed to eliminate lower quality bases.
- 2) To ensure high confidence alignments, only reads with robust alignments were considered for quantification. Furthermore, to reduce noise in the data, genes, transcripts, and splicing sites with fewer than 10 supporting reads were filtered out.
- 3) Additionally, while we have not performed additional RT-qPCR for validating the transcript isoforms found in our study, numerous studies have found significant and robust concordance between RNA-seq and corresponding RT-qPCR results, as explained efficiently in a short article by Coenye (Biofilm, 2021).

*(*As described in the above review, Everaert et al., (Sci Rep, 2017) found significant gene expression correlations between RT-qPCR results and results from 5 distinct RNA-seq pipelines for 18,080 protein-coding genes. The genes that showed discordance of their expression between the methodologies comprised an average of 1.8% of the total. These genes were reproducible across datasets and they exhibited low expression values, shorter transcripts characterized by reduced no. of exons, and poor read quality.)*

6. Results section is a bit long (some of the information could be moved to the discussion).

We thank the reviewer for the insightful suggestion. We have made specific changes to the results to make it more concise.

7. Figure resolution will need to be improved for almost all figures, please see specific figure comments below.

We thank the reviewer for the suggestion, we have uploaded images at 600 dpi and improved the font sizes in the figures and made changes as recommended in the following comments.

Possibly, the figure resolution also got compromised during conversion to PDF.

Minor comments:

Line 38: "minimal overlap between the gene and transcript expressions" needs to be clarified a bit. This could imply that single nuclei and single cell sequencing was done. Also, please see my comment above about using different programs for this analysis, I recommend it was done only using Swish R.

We thank the reviewer for the suggestion, we have modified the phrase as following:

“Using bulk RNA-seq, our analysis revealed minimal overlap between differentially expressed gene and transcript isoform, with distinct expression pattern across disease severity.”

Line 142: "gene and transcript-level differential expression analysis", once again this should be clarified, as I also don't see where this distinction is made in the Figure 1 study design overview. It would be good to get this clarity how these analyses were separated in this figure, maybe add the software/programs used into Figure 1.

We are grateful for the suggestions. We have modified Figure 1, study design to distinguish between gene and transcript level differential expression analysis and improved upon the phrase used as suggested.

Table 1 can be moved to supplemental.

We are thankful for the reviewer perspective. Considering the recommendations from previous studies and the direct relevance of clinical comparisons to our results, we included these clinical comparisons for different disease severities in the main manuscript.

We take this opportunity to request the Reviewer to allow us retain this aspect within the main body of the manuscript.

Figure 2 resolution needs to be improved, especially panels A-C as the genes are labelled but they aren't readable with the size of each plot. Panel E text is very difficult to read, also is panel E necessary?

We appreciate the reviewer feedback and have implemented the suggestions accordingly. Specifically, we have enhanced the font size in the figures and improved the overall readability of the plots.

Panel E serves as a graphical representation summarizing the topics explored in the subsequent sections, offering an overview of the results. Thus, towards broad readership help to understand the results, we would prefer and like to retain it in this figure.

Line 234: "significant positive enrichment" p-value? This paragraph goes on to discuss all enriched pathways, how was this enrichment defined? By how many genes in a pathway or fold-changes? If based on GSEA, there should be associated p-values.

We acknowledge the reviewer's suggestion. We have provided more clarity to the results section in the revised submission.

Additionally, we would like to highlight that the same information is also mentioned in the methods section under the Gene set enrichment analysis header and complete information already provided in **Supplementary File 6**.

Figure 3. Please improve the resolution of this figure. All text should be readable, this includes the numbers on panels B and F. Is panel A necessary, it seems like this just shows what alternative splicing is and doesn't add to the results?

We thank the reviewer for the suggestion. We have improved the figure by increasing the font size. As suggested, we have removed panel A for improving readability.

Line 295-299: I don't think defining alternative splicing in the results is appropriate, some detail could be given in the introduction, but not necessary here.

We thank the reviewer for the suggestion. We have relocated the section from the results to the introduction.

Figure 4: Resolution of this figure could also be improved. It would be worth thinking about splitting these larger figures into different figures if the panels together are so small it creates difficulties reading them.

We thank the reviewer for the suggestions, we have improved the figure resolution to 600dpi to improve readability in the revised submission.

Figure 5. Resolution of this figure needs to be improved; panel A could possibly be removed to increase the size of the other panels to improve readability.

We thank the reviewer for the suggestion. We have rearranged the panel to improve the readability of the text. We request to allow us to retain panel A for clarity of the result subsections in Figure 5.

Figure 6. The readability/resolution of this figure needs to be improved.

We acknowledge the reviewer's comment. We have increased the resolution of images to 600 dpi as well as split the panel and added part of it as **Supplementary Figure S2** to improve clarity.

Figure 7. This figure should be supplemental.

We have shifted Figure 7 to supplementary as **Supplementary Figure S3**.

Reviewer #2 (Comments to the Authors (Required)):

The authors collected blood samples from 112 NS1 antigen positive hospital admitted Dengue patients and classified them into three groups: Dengue without warning sign referred to as Mild (n=45), Dengue with warning sign which was further divided into two groups: Moderate (n=46) and Severe (n=21). The authors employed bulk RNA-seq analysis of whole blood samples to examine transcriptomic patterns associated with the disease severity. In particular, the authors focused on the analysis of alternative splicing and used multiple published bioinformatics approaches.

We sincerely acknowledge and appreciate your summary on our study. Your feedback is invaluable in improving the quality and rigor of our research. Your insightful assessment of our manuscript is greatly valued. We have provided a detailed response to all the suggestions below.

However, there is a lack of evidence indicating that the isoform level analysis using the obtained bulk RNA-seq data was reliable. Further, the isoform level analysis could be complicated by using total RNAs, as it would capture many immature transcripts as well. It was unclear how well the blood samples were stored either. Rigorous validations should be provided. Also, it is unclear if some observed changes in the isoform usage were simply due to the changes in cell types in the blood samples.

We acknowledge the reviewer's concerns. We have tried the best possible means for sample handling, storage, sequencing library preparation, sequencing and downstream bioinformatics analysis. We take this opportunity to share some of the specific details towards that.

- There have been prior studies utilizing bulk RNA sequencing to investigate alternative splicing patterns and the variance in transcript isoform usage. For instance, Kim et al.

(<https://doi.org/10.1016/j.celrep.2022.110341>), elucidated changes in the host immune response by examining gene expression patterns, splicing, and transcript isoform usage in individuals vaccinated with the DENV-specific candidate vaccine. They employed whole transcriptome sequencing of whole blood samples for their analysis. Aligning with our results, this study also found larger effect sizes of differentially expressed transcripts as compared to differentially expressed genes. They used LeafCutter tool to identify splicing variants and alterations in splicing isoform patterns. Their results were validated with RT-PCR which confirms the robusticity of the bioinformatic tools used to identify these spliced isoforms.

- A recent study by Lee et al., (<https://doi.org/10.1016/j.isci.2024.109177>), also used bulk RNA-sequencing datasets of whole blood samples from 190 individuals to identify differential alternative splicing events and variants associated with variation in host response to COVID-19 vaccination and SARS-CoV-2 variants.
- Moreover, Nakanishi et al. (<https://doi.org/10.1038/s41467-023-41912-4>), successfully identified spliced isoform variants of genes predictive of COVID-19 susceptibility and severity using bulk RNA-sequencing datasets of whole blood and lung tissue samples. The aforementioned studies collectively confirm that utilizing whole transcriptome analysis of whole blood samples is effective and reliable for evaluating the dynamics of alternative splicing and isoform usage as determinants of host's response to infection.
- We have strictly adhered to protocols regarding collection of blood samples, isolation and storage of RNA to maintain the integrity of the samples. Trained paramedical personnel at Max Hospital collected blood samples in EDTA-coated vials. The samples were stored at 4°C, and transported with cool packs to our Lab within 3-4 hours as both are within the same city of Delhi. RNA isolation was performed using freshly collected blood, followed by quantification of RNA concentration and purity using a Nanodrop. The isolated RNA was then stored at -80°C until library preparation. We have also ensured retaining high quality reads (Phred score >30) and employing robust cutoffs at each step to filter out genes/transcripts with <10 reads mapping to them. The significance testing for subsequent analysis was also stringent to effectively mitigate any potential artifacts in the results and interpretation. The results from our current study strengthen many findings

from previous studies conducted in various disease contexts or vaccination statuses, thereby validating the experimental methodology used.

- Based on one of your suggestions, we conducted cell type analysis using xCell (<https://doi.org/10.1186/s13059-017-1349-1>) to identify cell types in blood bulk RNA-seq. While we didn't observe significant differences between naive B cells, CD4+ and CD8+ T cells, variations were noted between macrophages and neutrophils (as shown below). These distinctions could be linked to the host immune response. We acknowledge that bulk RNA-seq may not be optimal for elucidating cell type-specific changes, single-cell-based studies will provide additional insights into the inferences highlighted from our research. Therefore, we've acknowledged this as a limitation and a potential area for future investigation in our study.

[Figure removed by editorial staff per authors' request]

Differential cell type abundance between mild, moderate and severe dengue patients based on gene count matrix.

May 21, 2024

RE: Life Science Alliance Manuscript #LSA-2024-02683-TR

Dr. Rajesh Pandey
Institute of Genomics and Integrative Biology
Mall Road, Delhi
Delhi 110007
India

Dear Dr. Pandey,

Thank you for submitting your revised manuscript entitled "Reduced Protein-Coding Transcript Diversity in Severe Dengue emphasize role of Alternative Splicing". We would be happy to publish your paper in Life Science Alliance pending final revisions necessary to meet our formatting guidelines.

- please be sure that the authorship listing and order is correct
- please be sure that all authors are mentioned in the Authors Contribution section in the manuscript file
- please move your main, supplementary figure, and table legends to the main manuscript text after the references section
- please add a callout for Figure 5E to your main manuscript text
- please remove the legend from the supplementary figures. Their captions should only appear in the manuscript file.
- the supplemental files should be relabeled as Supplemental Tables, and referred to as such in the manuscript

A. FINAL FILES:

B. MANUSCRIPT ORGANIZATION AND FORMATTING:

Sincerely,

May 22, 2024

RE: Life Science Alliance Manuscript #LSA-2024-02683-TRR

Dr. Rajesh Pandey
Institute of Genomics and Integrative Biology
Mall Road, Delhi
Delhi 110007
India

Dear Dr. Pandey,

Thank you for submitting your Resource entitled "Reduced Protein-Coding Transcript Diversity in Severe Dengue emphasize role of Alternative Splicing". It is a pleasure to let you know that your manuscript is now accepted for publication in Life Science Alliance. Congratulations on this interesting work.

DISTRIBUTION OF MATERIALS:

Again, congratulations on a very nice paper. I hope you found the review process to be constructive and are pleased with how the manuscript was handled editorially. We look forward to future exciting submissions from your lab.

Sincerely,
